# Tissue-specific multi-omics analysis of atrial fibrillation

Ines Assum 1,2,12, Julia Krause3,4,12, Markus O. Scheinhardt5, Christian Müller3,4, Elke Hammer6,7, Christin S. Börschel4,8, Uwe Völker 6,7, Lenard Conradi9, Bastiaan Geelhoed4,8,10, Tanja Zeller3,4,12, Renate B. Schnabel 4,8,12✉ & Matthias Heinig 1,2,11,12✉

Genome-wide association studies (GWAS) for atrial fibrillation (AF) have uncovered numerous disease-associated variants. Their underlying molecular mechanisms, especially consequences for mRNA and protein expression remain largely elusive. Thus, refined multi-omics approaches are needed for deciphering the underlying molecular networks. Here, we integrate genomics, transcriptomics, and proteomics of human atrial tissue in a cross-sectional study to identify widespread effects of genetic variants on both transcript (*cis*-eQTL) and protein (*cis*-pQTL) abundance. We further establish a novel targeted *trans*-QTL approach based on polygenic risk scores to determine candidates for AF core genes. Using this approach, we identify two *trans*-eQTLs and five *trans*-pQTLs for AF GWAS hits, and elucidate the role of the transcription factor NKX2-5 as a link between the GWAS SNP rs9481842 and AF. Altogether, we present an integrative multi-omics method to uncover *trans*-acting networks in small datasets and provide a rich resource of atrial tissue-specific regulatory variants for transcript and protein levels for cardiovascular disease gene prioritization.

[1] Computational Health Center, Helmholtz Zentrum München, Deutsches Forschungszentrum für Gesundheit und Umwelt (GmbH), München, Germany. [2] Department of Informatics, Technical University Munich, München, Germany. [3] University Center of Cardiovascular Science, University Heart and Vascular Center Hamburg, Hamburg, Germany. [4] Partner site Hamburg/Kiel/Lübeck, DZHK (German Center for Cardiovascular Research), Hamburg, Germany. [5] Institute of Medical Biometry and Statistics, University of Lübeck, University Hospital of Schleswig-Holstein, Lübeck, Germany. [6] Interfaculty Institute for Genetics and Functional Genomics, University Medicine Greifswald, Greifswald, Germany. [7] Partner site Greifswald, DZHK (German Center for Cardiovascular Research), Greifswald, Germany. [8] Department of Cardiology, University Heart and Vascular Center Hamburg, Hamburg, Germany. [9] Department of Cardiovascular Surgery, University Heart and Vascular Center Hamburg, Hamburg, Germany. [10] Department of Cardiology, University of Groningen, University Medical Center Groningen, Groningen, Netherlands. [11] Partner site Munich, DZHK (German Center for Cardiovascular Research), Munich, Germany. [12] These authors contributed equally: Ines Assum, Julia Krause, Tanja Zeller, Renate B. Schnabel, Matthias Heinig. ✉email: r.schnabel@uke.de; matthias.heinig@helmholtz-muenchen.de

Genome-wide association studies (GWAS) have discovered thousands of disease-associated single-nucleotide polymorphisms (SNPs) and improved our understanding of genetic and phenotypic relationships[1]. In this regard, GWAS have been applied to investigate atrial fibrillation (AF), which affects more than 30 million individuals worldwide[2]. More than 100 distinct genetic loci have been identified[3,4] and integrated into genome-wide polygenic risk scores (PRS) for AF risk prediction[5,6]. While rare monogenic effects exist, additive polygenic effects of many common variants explain a much higher proportion of AF risk in the population[7]. More than 95% of these GWAS variants are localized in non-coding regions[3] most likely acting through regulatory elements affecting expression of multiple genes and pathways that remain largely elusive[8].

A common approach to further investigate those mechanisms is to consider tissue-specific *cis*-acting expression quantitative trait loci (eQTL), where genetic variants affect the transcription of nearby genes. However, *cis*-eQTLs can only explain a fraction of the identified AF risk loci[3]. Therefore, *trans*-eQTLs, where the variant is distant to the target gene, and more complex genetic or epigenetic mechanisms need to be considered[8–10]. To date, it remains difficult to quantify the contribution of *cis*- and *trans*-variants to the heritability of complex diseases such as AF.

Recently, the contribution of *trans*-effects to the genetic architecture of complex polygenic traits was theoretically assessed by the omnigenic model[11]. Based on this model, it was estimated that *trans*-genetic effects explain at least 70% of the disease heritability by indirect propagation through gene regulatory networks[11]. Within these networks, multiple *trans*-effects can accumulate on just a few central genes, so-called core genes, which in turn are functionally related to a phenotype. Identifying those core genes by *trans*-eQTLs remains challenging due to the small effect size of each individual locus[9,12] and the associated large multiple testing burden. Since a PRS summarizes the genetic risk information, it can act as a proxy for the accumulation of *trans*-effects in one score[12]. By correlating the score with transcript expression (eQTS), the propagation of *trans*-effects to mRNA level[12] can be evaluated. However, not only transcript abundance, but also the abundance of translated proteins can determine phenotypic consequences. To date, little is known about genetic effects on protein levels (pQTLs), e.g. through post-transcriptional regulation[13–16], especially in atrial tissues. It is both challenging and important to establish methods to identify AF core genes and to integrate data from multiple omics levels to improve the understanding of genotype–phenotype relationships.

In this work, we present a multi-omics analysis that uses genomics, transcriptomics and proteomics of human atrial tissue to better understand how genetics are related to molecular changes in AF. The first aim was to systematically integrate omics data and identify genome-wide *cis*-regulatory mechanisms on transcript as well as protein level. We reasoned that core genes are key for a better understanding of complex molecular patho-mechanisms of AF. Therefore, the second aim was to identify candidate core genes for AF and the *trans*-acting regulatory networks that link them to AF GWAS loci. We develop an approach combining the correlation of gene expression with a PRS for AF[6,12] and pathway enrichment analysis to identify AF-associated biological processes. Based on those processes, candidate core genes for targeted *trans*-QTL analyses with AF GWAS SNPs are selected. This approach allows the identification of putative core genes, their molecular networks and downstream consequences in AF.

## Results

***Cis*-QTL analysis.** Microarray transcriptomics and mass spectrometry-based proteomics were used to profile human atrial tissue samples collected during coronary artery bypass surgery. We analyzed disease-independent effects of genetic variants on transcript, *cis*-expression quantitative trait loci (*cis*-eQTLs), and protein levels, *cis*-protein quantitative trait loci (*cis*-pQTLs), of nearby genes. All *cis*-QTLs were calculated using expression values for 16,306 genes and 1337 proteins (Table 1) with additional PEER[17,18] factors, an established method which allows to adjust for known as well as unknown confounders (see methods, Supplementary Fig. 1). Comparison of PEER factors and observed covariates showed that PEER factors successfully captured disease status, cardiovascular risk factors and technical covariates (Supplementary Fig. 2). We assessed the replication rate of our eQTLs in GTEx[19] v7 atrial appendage tissue. Effect sizes for the best eQTL ($P < 1 \times 10^{-5}$) per gene showed a correlation of 0.81 ($P = 2.3 \times 10^{71}$) in GTEx, 62% replicated (GTEx $P < 1 \times 10^{-5}$) and 87% showed concordant allelic effects (see also Supplementary Note 1 *Cis*-QTL replication[20], Supplementary Fig. 3). Furthermore, correlations between transcript and protein abundances were comparable to previous studies[14] (Supplementary Fig. 4, Supplementary Table 1) indicating high quality of the proteomics measurements.

***Cis*-regulatory patterns in atrial tissue.** We first sought to functionally characterize the *cis*-regulatory variants. Local genetic variation can lead to different modulations in mRNA and protein abundance, which are commonly attributed to transcriptional and post-transcriptional regulation[14,15,21]. Protein abundances are

## Table 1 Summary of tested data and discovered *cis*-quantitative trait loci.

**Results for all available transcriptomics and proteomics measurements**

| | Tested | | | FDR < 0.05: | | | $P < 1 \times 10^{-5}$ | | | |
|---|---|---|---|---|---|---|---|---|---|---|
| | SNPs | Pairs | Genes | Pairs | Genes | Loci | Pairs | Genes | Loci | *N* |
| eQTL | 4,861,118 | 56,139,851 | 16,306 | 57,403 | 1058 | 1657 | 40,267 | 552 | 870 | 75 |
| pQTL | 2,323,504 | 4,508,654 | 1337 | 4081 | 91 | 139 | 2543 | 45 | 71 | 75 |

**Results only for genes with both transcriptomics and proteomics measurements**

| | | | | | | | | | | |
|---|---|---|---|---|---|---|---|---|---|---|
| eQTL | 2,249,758 | 4,198,168 | 1243 | 4603 | 124 | 201 | 3218 | 64 | 109 | 75 |
| pQTL | 2,249,758 | 4,198,168 | 1243 | 3906 | 87 | 133 | 2406 | 42 | 66 | 75 |
| ratioQTL | 2,249,758 | 4,198,168 | 1243 | 563 | 16 | 23 | 575 | 18 | 27 | 66 |

Discovered *cis*-QTLs in human heart right atrial appendage tissue for mRNA and protein measurements. Two-sided *t*-tests were evaluated and Benjamini-Hochberg procedure to calculate FDR per omic was applied to account for multiple comparisons. Listed are the results for significance thresholds FDR < 0.05 and nominal *P*-value < $1 \times 10^{-5}$. Loci denote the number of independent loci derived by LD clumping. Source data are provided as a Source Data file.
*QTL* quantitative trait loci, *FDR* false discovery rate, *LD* linkage disequilibrium, *eQTL* expression quantitative trait loci, *pQTL* protein quantitative trait loci, *N* sample size.

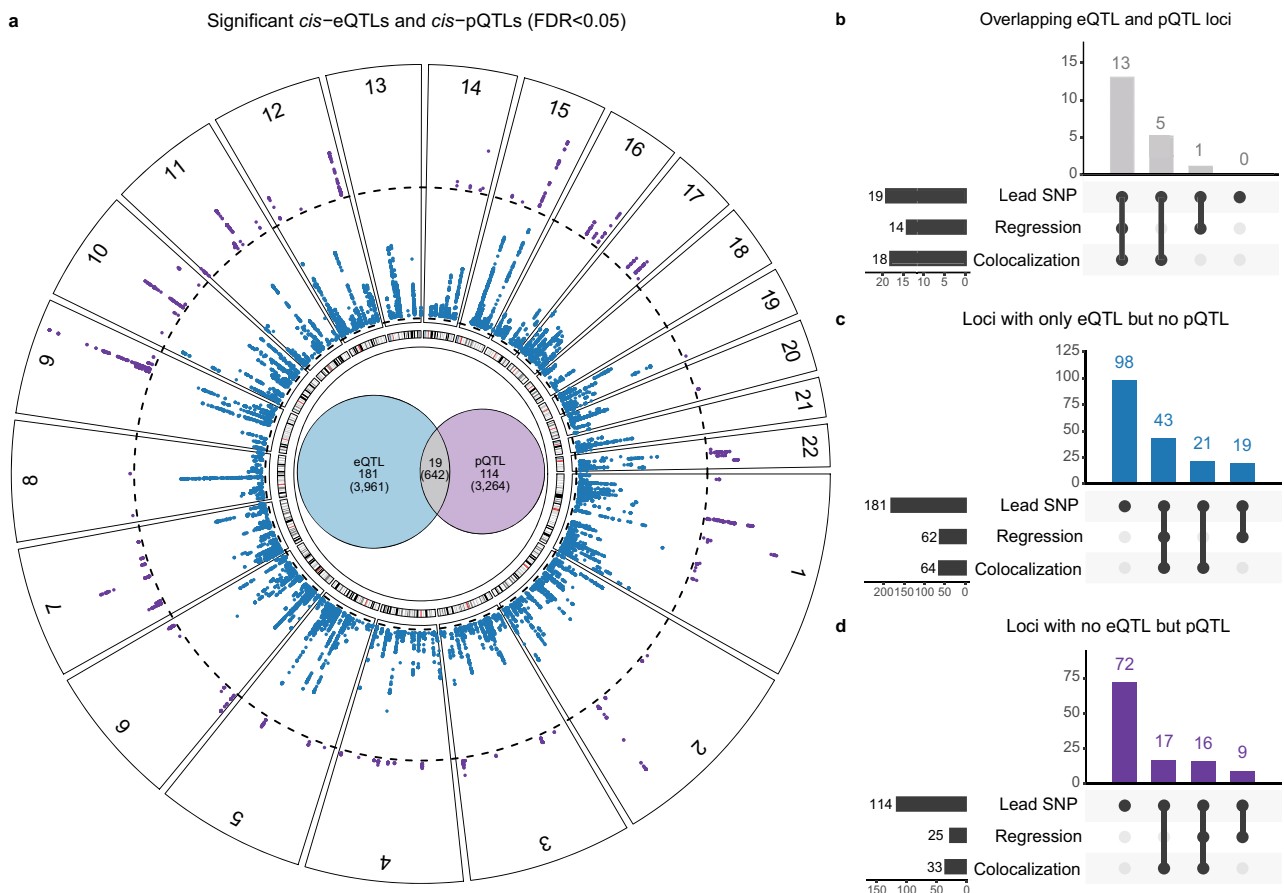

**Fig. 1 Significant *cis*-eQTLs, *cis*-pQTLs and their overlap. a** Circular plot of the significant *cis*-eQTLs (blue) and pQTLs (purple) at a FDR cutoff of 0.05 (dotted line, plot created using the R package circlize[85]). Considering only genes with both transcriptomics and proteomics measurements, we visualized the overlap of significant eQTLs and pQTLs in the circle center. In total, the lead SNP-gene pair of 200 QTL clumps in 124 genes had a significant eQTL and 133 loci in 87 genes a significant pQTL. Only 19 lead variants (13 genes) had an eQTL and pQTL for the same gene. The numbers in brackets represent the number of significant SNP-gene pairs. **b** Characterization of overlapping eQTL and pQTL loci. All 19 LD clumps (based on eQTL and pQTL summary statistics) where the lead SNP-gene-pair was a significant eQTL and pQTL were classified as a shared QTL by either our residual regression approach or colocalization analysis. **c** Characterization of eQTL loci without a corresponding pQTL. Only 83 out of 181 LD clumps (based on eQTL and pQTL summary statistics) that had a lead SNP-gene-pair with a significant eQTL but no pQTL were classified as an independent eQTL by either our residual regression approach or colocalization analysis. **d** Characterization of pQTL loci without a corresponding eQTL. Only 42 out of 114 LD clumps (based on eQTL and pQTL summary statistics) that had a lead SNP-gene-pair with a significant pQTL but no eQTL were classified as an independent pQTL by either our residual regression approach or colocalization analysis. eQTL expression quantitative trait loci, pQTL protein quantitative trait loci, QTL quantitative trait loci, FDR false discovery rate, LD linkage disequilibrium, SNP single-nucleotide polymorphism. Source data are provided as a Source Data file.

suggested as more direct determinants for phenotypic consequences of expression QTLs[14] emphasizing the need to integrate mRNA and proteomic measurements to better understand functional genotype–phenotype relationships. Thus, only for the following characterization of *cis*-genetic variation on transcripts and proteins, we focused on genes with both transcriptomics and proteomics measurements (1,243 genes, Table 1). As observed previously[14,15,21], significant eQTLs and pQTLs (FDR < 0.05) differ considerably. On gene level, 32% of the top eQTLs replicated in pQTLs and 50% vice versa, with a Pearson correlation of 0.58 and 0.66 for corresponding effect sizes (see Supplementary Information Overlap of *cis*- eQTLs and pQTLs). Considering all individual SNP-gene pairs, only 8.2% of significant associations are shared between mRNA and proteins (Fig. 1a, Supplementary Fig. 5, Supplementary Table 2). Linkage disequilibrium (LD) clumping was performed on eQTL and pQTL summary statistics simultaneously allowing for the partitioning of *cis*-regions into independent loci for further evaluation.

Divergence of mRNA from protein abundance can arise through diverse molecular mechanisms which we additionally analyzed by calculating protein-per-mRNA ratios (ratioQTLs)

(see methods). Lack of overlap can either indicate independent effects or lack of power in one of the studies. To distinguish these scenarios, we quantified for every variant if its effect was shared between the omic levels using a linear regression based residual QTL analysis. A residual eQTL considers the association between a variant and the mRNA measurements after removing variation shared with protein measurements for the same gene. Vice versa, a residual pQTL describes the association between the variant and protein levels after removing variation shared with mRNA. Having removed variation that was shared across omics levels for each gene, we can now assess the omic-specific effect of a variant. Three simple regulatory categories were defined by our regression approach (see methods, Fig. 2, Supplementary Fig. 6, Supplementary Table 3). We compared these categories to a formal colocalization analysis based on approximate Bayes Factor analysis[22] to estimate posterior probabilities for a shared or independent causal variant within each LD clump:

(i)  For 11 genes, 14 independent loci with shared *cis*- eQTL/ pQTL were identified, where the lead SNP is associated with

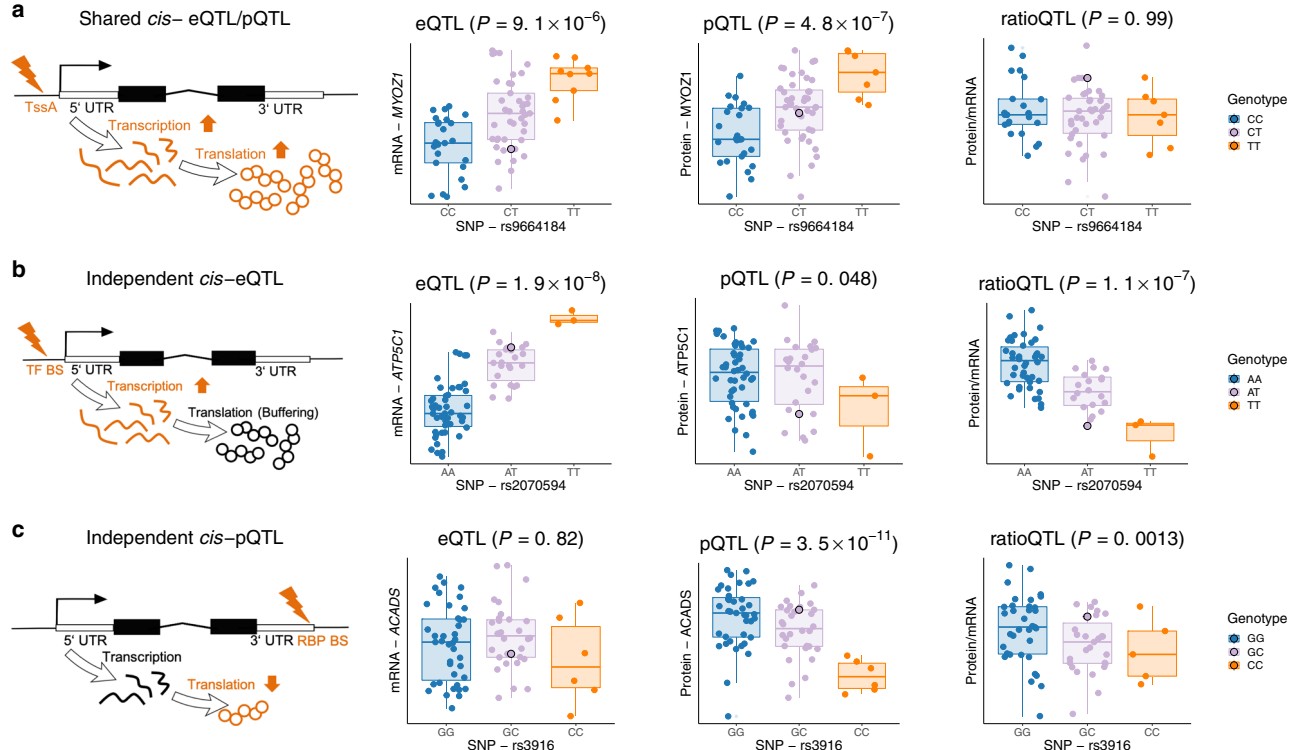

**Fig. 2 Different genetic regulatory patterns derived by multi-omics *cis*-QTL integration. a** Shared eQTLs/pQTLs represent QTLs, where the effect of transcriptional regulation translates into mRNA and protein abundance exemplified by the significant SNP-gene pair rs9664184-MYOZ1. No corresponding ratioQTL can be observed as the genetic variation is shared across both omics levels. **b** Independent eQTLs depict variants with regulation on mRNA but not on protein level displayed by the significant SNP-transcript pair rs2070594-ATP5C1. **c** Independent pQTLs represent variants that show regulation only on protein level as shown for the SNP-protein pair rs3916-ACADS. Genetic influence is not observable on transcript level. In the boxplots, the lower and upper hinges correspond to the first and third quartiles (the 25th and 75th percentiles). The median is denoted by the central line in the box. The upper/lower whisker extends from the hinge to the largest/smallest value no further than 1.5 × IQR from the hinge. Nominal *P*-values were derived based on two-sided *t*-tests for *N* = 75 (eQTLs), *N* = 75 (pQTL) and *N* = 66 (ratioQTL) biologically independent samples. To assess significance, FDR correction per omic based on the Benjamini-Hochberg procedure was applied to account for multiple comparisons. eQTL expression quantitative trait loci, pQTL protein quantitative trait loci, ratioQTL ratio quantitative trait loci, TssA active transcription start site, UTR untranslated region, TF BS transcription factor binding site, RBP RNA binding protein, SNP single-nucleotide polymorphism, IQR interquartile range. Source data are provided as a Source Data file.

both mRNA expression and the respective protein abundance as depicted in Fig. 2a. The corresponding variants were primarily enriched in *cis*-regulatory elements such as active transcription start sites (TssA) (Supplementary Fig. 6a, Supplementary Fig. 7). For all but one loci, colocalization analysis also confirmed a shared causal variant (Fig. 1b).

(ii) For 37 genes, 62 independent loci with independent eQTLs were identified, where the lead SNP is associated only with transcript levels, but the respective protein abundance is independent of the genotype (Fig. 2b). Corresponding variants were enriched in elements regulating transcription, e.g. transcription factor binding site (TF BS) or enhancer regions, and within splicing regions (Supplementary Fig. 6b, Supplementary Fig. 7a). Possible mechanisms involved in unchanged protein levels remain largely elusive and range from adaptation of translational rate to protein degradation and long-non-coding RNAs[23,24]. For 43 out of 62 LD clumps, colocalization analysis also confirmed an independent eQTL (Fig. 1c).

(iii) For 21 genes, 25 independent loci with independent pQTLs were identified, where the lead SNP affects only protein abundance (Fig. 2c). pQTL variants were enriched for exonic regions and although not significantly, in binding sites of RNA binding proteins (RBP) (Supplementary

Fig. 6c, Supplementary Fig. 7a), where they may influence mRNA translation resulting in an independent pQTL association[25]. For 16 out of 25 LD clumps, colocalization analysis also confirmed an independent pQTL (Fig. 1d).

For only a small fraction of 5 out of 181 loci with an eQTL but no pQTL, colocalization analysis suggested a shared QTL and the other way around, 7 out of 114 loci with a pQTL but no eQTL may be actually shared. Altogether, we confirmed that QTL variants corresponding to different categories tended to cluster in distinct genomic regions (Supplementary Fig. 7, Supplementary Note 1)[14,18,26]. By integrating matched transcriptome and proteome data, we were able to differentiate functional regulatory mechanisms not observable by transcriptomics only.

*GWAS overlap and enrichment.* In order to investigate genotype–phenotype relationships in the context of cardiovascular disease, we used all available *cis*-QTL data (not only those quantified on both omics levels, FDR < 0.05) to annotate GWAS variants for phenotypes either related to cardiovascular measurements or cardiovascular disease (Fig. 3a).

Of all the overlaps between GWAS hits and *cis*-QTLs (Supplementary Table 4), AF-related loci were most abundant (17 eQTLs, four also with pQTL, see Supplementary Table 5). Furthermore, we found an independent pQTL overlapping with the GWAS hit for creatine kinase levels (Fig. 3b). This genetic

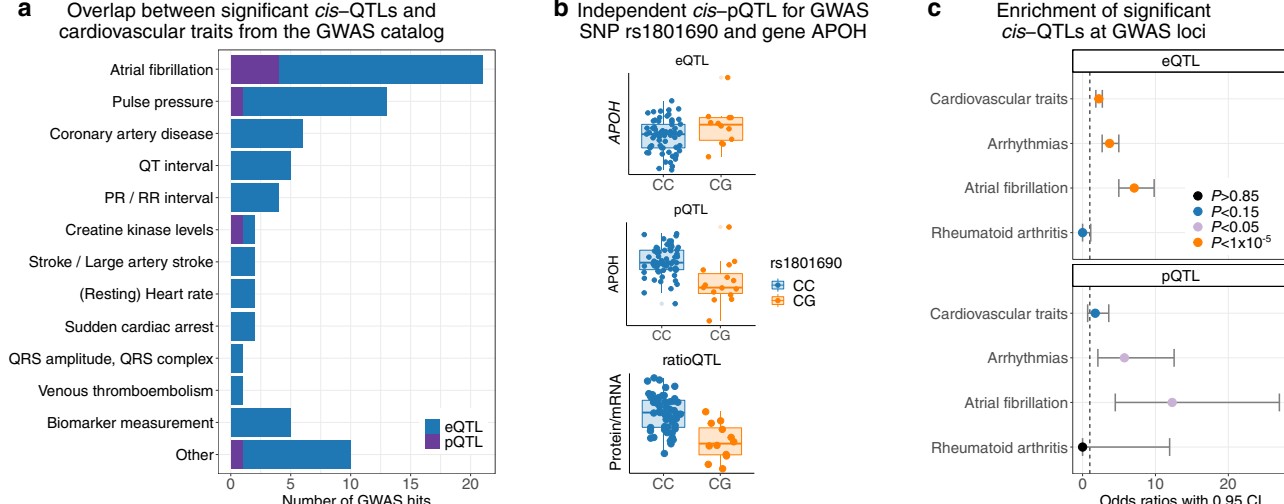

**Fig. 3 Overlap of *cis*-QTL associations with GWAS hits annotated in the GWAS catalog. a** Overview of significant *cis*- eQTLs and pQTLs (FDR < 0.05) overlapping with GWAS hits for different disease traits. **b** Independent pQTL for GWAS hit creatine kinase levels. Shown are the non-significant *cis*-eQTL and the significant *cis*- pQTL and ratioQTL for the SNP rs1801690 and the gene APOH (FDR < 0.05). Statistics were derived based on two-sided *t*-tests for $N = 75$ (eQTLs), $N = 75$ (pQTL) and $N = 66$ (ratioQTL) biologically independent samples. A FDR < 0.05 per omic based on the Benjamini-Hochberg procedure was applied to assess significance and to account for multiple comparisons. In the boxplots, the lower and upper hinges correspond to the first and third quartiles (the 25th and 75th percentiles). The median is denoted by the central line in the box. The upper/lower whisker extends from the hinge to the largest/smallest value no further than 1.5 × IQR from the hinge. **c** For three different trait categories (cardiovascular traits, arrhythmias and atrial fibrillation) as well as rheumatoid arthritis as a negative control, the enrichment of GWAS hits at significant *cis*-QTLs (FDR < 0.05) was evaluated. Enrichments were calculated using Fisher's exact test (two-sided). 4,815,266 (eQTL) and 2,301,873 (pQTL) SNPs were evaluated for 7,817/4,661 (eQTL/ pQTL) cardiovascular trait, 2,287/1,006 (eQTL/pQTL) arrhythmic, 691/394 (eQTL/pQTL) AF and 468/297 (eQTL/pQTL) RA GWAS hits. Odds ratios are presented with their 95% CI. Source data are provided as a Source Data file. QTL quantitative trait loci, GWAS genome-wide association study, SNP single-nucleotide polymorphism, eQTL expression quantitative trait loci, pQTL protein quantitative trait loci, ratioQTL ratio quantitative trait loci, CI confidence interval, IQR interquartile range. Source data are provided as a Source Data file.

effect was not detected on mRNA level illustrating the importance of proteomics data. In addition, we systematically assessed whether significant QTLs are enriched at GWAS loci in the hierarchical groups cardiovascular traits, arrhythmias and AF (two-sided Fishers exact test). We identified a strong significant overrepresentation ($P < 1 \times 10^{-5}$) of eQTLs at GWAS hits for all three groups, and a significant overrepresentation ($P < 0.05$) of pQTLs in variants annotated with arrhythmias and AF (Fig. 3c). Additionally, we evaluated rheumatoid arthritis (RA) as negative control, a trait that should not share atrial-specific disease mechanisms. Indeed, there were no overlaps between *cis*-QTLs and RA GWAS loci. Alltogether, we presented widespread effects of *cis*-acting variants on gene expression and protein abundance in atrial tissue and a possible relation to AF.

***Trans*-QTL analysis.** We further extended *cis*-regulatory analyses by investigating *trans*-effects. Specifically, we addressed a key hypothesis of the omnigenic model[11], which postulates the existence of core genes. Core genes are central genes with *trans*-associations to AF GWAS loci, whose expression levels directly affect the disease phenotype. Here we sought to identify candidate core genes for AF to understand the contribution of *trans*-genetic effects in the pathology of AF. To prioritize genes satisfying the properties predicted by the omnigenic model, we evaluated the accumulation of *trans*-effects, their relevance in gene regulatory networks, and the association with AF by the following strategy (Fig. 4):

(1) We evaluated the cumulated *trans*-effects of AF-associated variants on expression by ranking genes based on their correlation of mRNA and protein abundance with the PRS for AF[6] (see Supplementary Fig. 8), so called expression/ protein quantitative trait score (eQTS/pQTS)[12]. While correcting for possible *cis*-effects by including the top

SNP per independent *cis*-QTL loci, the PRS served as a proxy for an aggregation of AF-related *trans*-effects across the whole genome.

(2) To identify genes sharing molecular function and representing biological networks that propagate *trans*-effects to core genes, gene set enrichment analysis (GSEA)[27,28] was performed on the eQTS and pQTS rankings. Genes driving the enrichment of multiple gene sets were selected as core gene candidates.

(3) The link between the core gene candidates and AF was established based on a significant *trans*-eQTL or pQTL for an AF GWAS hit and further supported by differential protein abundance analysis.

*Identification of candidate AF core genes.* Based on this strategy, we first used the GO biological process gene set annotations, which are not a priori disease-related, to recover processes functionally related to AF. Using the eQTS ranking of all measured transcripts (top hits in Supplementary Table 6) as background for the gene set enrichment, 81 GO biological processes were enriched (adjusted *P*-value < 0.05, Supplementary Table 7) mostly related to heart muscle or energy metabolism, including the processes GO:0006091 (generation of precursor metabolites and energy), GO:0055117 (regulation of cardiac muscle contraction), GO:0048738 (cardiac muscle tissue development) and GO:0002027 (regulation of heart rate). Furthermore, we identified three processes that implicate calcium homeostasis (GO:0010880 regulation of release of sequestered calcium ion into cytosol by sarcoplasmic reticulum, GO:0010881 regulation of cardiac muscle contraction by regulation of the release of sequestered calcium ion, and GO:0010882 regulation of cardiac muscle contraction by calcium ion signaling). For pQTS rankings (top hits in

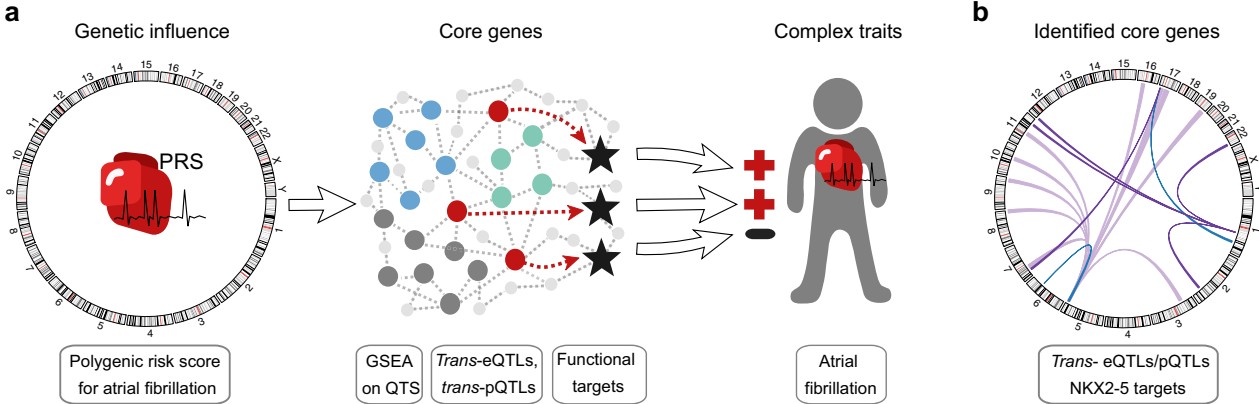

**Fig. 4 Graphical illustration of the strategy for *trans*-QTL analysis to identify AF-relevant genes. a** Overview: Based on patient-specific PRS values for AF correlated with transcript and protein expression, we performed GSEA to preselect genes for *trans*-eQTL and pQTL analyses from the leading edge of enriched pathways. Core genes were identified as significant *trans*-eQTLs or *trans*-pQTLs. We further assessed their functional targets to investigate the genotype–phenotype relationship in the context of AF. Graphical concept adapted from Liu et al.[11]. **b** Identified core genes as *trans*-eQTLs (blue), *trans*-pQTLs (purple) (FDR < 0.2) and functional NKX2-5 targets (light purple). PRS, genome-wide polygenic risk score; GSEA gene set enrichment analysis, QTS quantitative trait score, eQTL expression quantitative trait loci, pQTL protein quantitative trait loci, FDR false discovery rate, AF atrial fibrillation, blue, green or gray dots = core gene candidates, red dots = core genes with *trans*- eQTL/pQTL, stars = functional targets of core genes. Circular plots were created with the R package circlize[85].

**Table 2 *Trans*-QTL results.**

| Variant | | | | Gene | | *Trans*-QTL | | | | |
|---|---|---|---|---|---|---|---|---|---|---|
| GWAS SNP | QTL SNP | Chr | Position | Symbol | Chr | QTL | β | T value | P-value | FDR |
| rs9675122 | rs11658168 | chr17 | 7,406,134 | TNNT2[a,d] | chr1 | Transcript | −0.517 | −4.27 | 6.43 × 10⁻⁵ | 0.081 |
| rs9481842 | rs9481842 | chr6 | 118,974,798 | NKX2-5[b] | chr5 | Transcript | −0.593 | −4.27 | 6.54 × 10⁻⁵ | 0.081 |
| rs34292822 | rs11588763 | chr1 | 154,813,584 | CYB5R3 | chr22 | Protein | −0.786 | −4.89 | 6.86 × 10⁻⁶ | 0.113 |
| rs34292822 | rs11588763 | chr1 | 154,813,584 | NDUFB3[c] | chr2 | Protein | −0.916 | −4.44 | 3.56 × 10⁻⁵ | 0.133 |
| rs9675122 | rs11658168 | chr17 | 7,406,134 | HIBADH | chr7 | Protein | −0.512 | −4.43 | 3.66 × 10⁻⁵ | 0.133 |
| rs34292822 | rs11588763 | chr1 | 154,813,584 | NDUFA9[d] | chr12 | Protein | −0.752 | −4.42 | 3.85 × 10⁻⁵ | 0.133 |
| rs34292822 | rs11588763 | chr1 | 154,813,584 | DLAT | chr11 | Protein | −0.716 | −4.40 | 4.05 × 10⁻⁵ | 0.133 |

Significant *trans*-eQTLs and pQTLs for a FDR < 0.2 (Benjamini-Hochberg procedure to account for multiple comparisons per omic). Two-sided *t*-tests were performed on 23 transcripts with 74 samples and 152 proteins with 73 samples of human heart right atrial appendage tissue for 108 variants associated with atrial fibrillation from the GWAS catalog (or their proxy, if the GWAS SNP was not measured). Calculations were carried out using the SNP rs11658168 as a proxy for the GWAS SNP rs9675122 as well as rs11588763 instead of the GWAS SNP rs34292822.
*eQTL* expression quantitative trait loci, *pQTL* expression quantitative trait loci, *QTL* quantitative trait loci, *FDR* false discovery rate, *GWAS* genome-wide association study, *SNP* single-nucleotide polymorphism, *Chr* Chromosome.
[a]Mutation known to affect cardiovascular phenotypes.
[b]Mutation known to affect arrhythmias.
[c]Differential expression functional impairment for cardiovascular phenotypes.
[d]Differential expression or functional impairment for arrhythmias; For details to disease links in literature see Supplementary Table 12. Source data are provided as a Source Data file.

Supplementary Table 8) and restricting the background only to those proteins quantified in our dataset, one GO biological process (GO:0044281 small molecule metabolic process) connected to metabolism was enriched (adjusted *P*-value < 0.05, Supplementary Table 9).

Our pathway enrichment approach yielded 23 transcripts (Supplementary Table 10) and 152 proteins (Supplementary Table 11) as core gene candidates that we used to calculate *trans*-QTLs with 108 AF GWAS SNPs. On mRNA level, we identified two *trans*-eQTLs encoding for a cardiac structural protein (rs11658168-*TNNT2*) and a transcription factor (rs9481842-*NKX2-5*) (Table 2). Since NKX2-5 was not detected on protein level using mass spectrometry, we performed additional Western blot experiments to identify the respective pQTL (see Supplementary Fig. 9-10, $\beta = -0.45$, $T = -2.9$, $P = 0.049$, $N = 29$, df = 21, two-sided *t*-test). On protein level, we discovered five *trans*-pQTLs which are all connected to metabolism (rs11588763-CYB5R3/NDUFB3/NDUFA9/DLAT, rs11658168-HIBADH) (Table 2). Noticeably, four out of five identified genes encode for mitochondrial enzymes (HIBADH) or enzyme subunits

(NDUFA9, NDUFB3, DLAT). More than half of the putative core genes have already been mentioned by other studies in the context of arrhythmias and other cardiovascular diseases[29–35] (detailed findings see Supplementary Table 12 and differential expression results[30,36] in Supplementary Table 13-14) which independently replicates the disease link.

*NKX2-5 transcription factor network.* In order to get more detailed information about complex molecular mechanisms underlying AF, we further analyzed the TF network of *NKX2-5* (see Fig. 5a) since the TF has already been described in the context of cardiac development[37], AF[31,32,38] and congenital heart diseases[39].

To evaluate the downstream effects of the SNP rs9481842 via the TF NKX2-5 in AF, we analyzed the influence of *NKX2-5* transcript levels on target genes by estimating NKX2-5 TF activity (TFA). We annotated NKX2-5 binding sites in promoter regions based on published iPSC-derived cardiomyocyte ChIP-seq and promoter-capture HiC data to identify almost 10,000 target genes. The number of binding sites per gene in open chromatin regions were

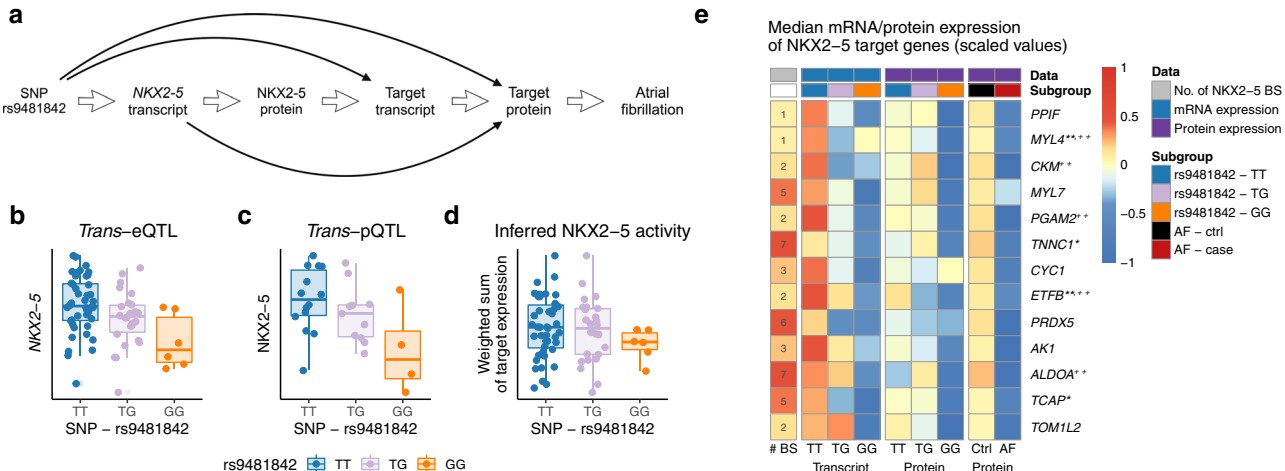

**Fig. 5 NKX2-5 activity controlled by AF GWAS variant rs9481842. a** Graphical illustration of NKX2-5 TF target gene analysis in AF. **b** Strong *trans*-eQTL of the SNP rs9481842 with the *NKX2-5* transcript for $N = 75$ independent biological samples. **c** Validation of the NKX2-5 *trans*-eQTL on protein level (*trans*-pQTL) using western blot analysis in remaining tissue samples ($N = 29$ independent biological samples). **d** NKX2-5 activity estimation based on target mRNA expression stratified by the rs9481842 genotype for $N = 75$ independent biological samples. **e** Depicted are functional NKX2-5 targets with the number of TF binding sites (column 1), *trans*-eQTL strength (columns 2–4), *trans*-pQTL strength (columns 5–7) and protein level in AF (columns 8–9). The colour scale represents median transcript or protein values per group (=columns). Residuals corrected for fibroblast-score and RIN-score / protein concentration with subsequent normal-quantile-normalization per gene were used to calculate the medians per group. A quantitative description of the qualitative results presented in the heatmap can be found in Supplementary Table 16 and Table 3. In the boxplots, the lower and upper hinges correspond to the first and third quartiles (the 25th and 75th percentiles). The median is denoted by the central line in the box. The upper/lower whisker extends from the hinge to the largest/smallest value no further than 1.5 × IQR from the hinge. AF atrial fibrillation, QTL quantitative trait loci, BS binding site, IQR interquartile range *Mutation known to affect cardiovascular phenotypes, **Mutation known to affect arrhythmias, +Differential expression or functional impairment for cardiovascular phenotypes, ++Differential expression or functional impairment for arrhythmias. Source data are provided as a Source Data file.

counted for each gene and the TFA was computed as the sum of target transcript expression weighted by the number of binding sites. We observed a high correlation between the SNP rs9481842 and the *NKX2-5* transcript (cor = −0.43, $P = 1.4 \times 10^{-4}$, two-sided Pearson's correlation, Fig. 5b, Supplementary Fig. 11a) as well as for the direct molecular link between the *NKX2-5* transcript and the TF activity (cor = 0.36, $P = 1.3 \times 10^{-4}$, one-sided Pearson's correlation, Supplementary Fig. 11c). In addition, there was a weak association between SNP rs9481842 and the TF activity (cor = −0.13, $P = 0.145$, one-sided Pearson's correlation, Fig. 5d, Supplementary Fig. 11b) most likely attributed to the indirect link through *NKX2-5*. Partial correlation analysis further supported NKX2-5 being the causal link between SNP and target expression (Supplementary Table 15). Based on the western blot data that was used to identify the NKX2-5 pQTL (Fig. 5c), we also observed a high correlation (Spearman's rank correlation $\rho = 0.42$, $P = 0.026$) between the calculated TF activity and the actual protein abundance (Supplementary Fig. 12).

Next, to elucidate the role of NKX2-5 as a link between the disease variant and AF, we further analyzed its effect on specific targets, which we also prioritized as putative core genes. Overall, we identified 13 functional targets that are significantly influenced by the SNP rs9481842 as well as *NKX2-5* transcript levels on both mRNA and protein level (see methods and Supplementary Fig. 13 for details, Supplementary Table 16). For these 13 targets, we observed a consistent downregulation on mRNA and protein level with respect to the rs9481842 risk allele (Fig. 5e). As the core gene model predicts a direct effect of core gene expression on the phenotype[11], we evaluated the protein abundance of the NKX2-5 target genes in patients with AF compared to patients in sinus rhythm to assess functional connection to the disease. For all targets, AF cases showed lower protein levels (Fig. 5e). When adjusting for common risk factors of AF, five out of 13 targets

showed a nominal *P*-value $P < 0.05$ (Table 3). More importantly, the identified target set collectively displayed a strong association with AF on proteomics level (GSEA $P = 7.17 \times 10^{-5}$). This serves as independent validation of the disease link, since these genes were identified based on molecular data in our cohort in combination with public AF annotations without using the actual cohort phenotypes.

The coordinated downregulation of those 13 targets in patients with AF was replicated in two independent datasets (Fig. 6a, for details see methods) with a GSEA *P*-value of 0.00593 for RNA-seq data in right atrial appendage samples of GSE128188[36], a GSEA *P*-value of 0.0248 for RNA-seq data in left atrial appendage samples of GSE128188 (see Supplementary Table 13 for differential expression results) and with a GSEA *P*-value of $2.43 \times 10^{-3}$ in left atrial samples from the proteomics dataset PXD006675[30] (see Supplementary Table 14 for differential protein abundance results). Similarly, regulation of our identified targets by the TF NKX2-5 was corroborated by coexpression of *NKX2-5* mRNA with target transcript levels in GTEx atrial appendage tissue and GSE128188 (Fig. 6b).

Furthermore, the majority of the identified proteins are in fact involved in contractile function (MYL4, MYL7, TNNC1 and TCAP) or metabolism (PPIF, CKM, AK1, PGAM2, CYC1, ETFB and ALDOA), two mechanisms linked to processes involved in the pathophysiology of AF.

At this point, our identified putative core genes point to potential novel targets for further experimental research to better understand molecular consequences of genetics underlying AF.

## Discussion

We present a comprehensive multi-omics analysis that integrates genomics, transcriptomics and proteomics in human atrial tissue in a case control cohort of AF. This unique dataset allowed us to

**Table 3 Putative core genes and functional targets with disease association.**

| | | | Protein AF association | | | |
|---|---|---|---|---|---|---|
| Gene | Chr | Type | $\beta$ | T-value | P-value | FDR |
| TNNT2[a,d] | chr1 | Trans-eQTL | −0.0609 | −1.61 | 0.113 | 1.00 |
| NKX2-5[b] | chr5 | Trans-eQTL | | | | |
| CYB5R3 | chr22 | Trans-pQTL | −0.0212 | −0.662 | 0.511 | 1.00 |
| NDUFB3[c] | chr2 | Trans-pQTL | −0.0631 | −1.35 | 0.182 | 1.00 |
| HIBADH | chr7 | Trans-pQTL | −0.0454 | −1.24 | 0.218 | 1.00 |
| NDUFA9[d] | chr12 | Trans-pQTL | −0.0533 | −1.20 | 0.235 | 1.00 |
| DLAT | chr11 | Trans-pQTL | −0.0231 | −0.579 | 0.564 | 1.00 |
| PPIF | chr10 | NKX2-5 target | −0.0342 | −1.13 | 0.261 | 1.00 |
| MYL4[b,d] | chr17 | NKX2-5 target | −0.0270 | −0.664 | 0.509 | 1.00 |
| CKM[d] | chr19 | NKX2-5 target | −0.0875 | −2.78 | 0.00705 | 0.120 |
| MYL7 | chr7 | NKX2-5 target | −0.0421 | −1.04 | 0.304 | 1.00 |
| PGAM2[d] | chr7 | NKX2-5 target | −0.175 | −3.70 | 0.000452 | 0.00813 |
| TNNC1[a] | chr3 | NKX2-5 target | −0.0557 | −1.71 | 0.0929 | 1.00 |
| CYC1 | chr8 | NKX2-5 target | −0.0946 | −2.14 | 0.036 | 0.545 |
| ETFB[b,d] | chr19 | NKX2-5 target | −0.0553 | −1.65 | 0.105 | 1.00 |
| PRDX5 | chr11 | NKX2-5 target | −0.0524 | −1.79 | 0.0789 | 1.00 |
| AK1 | chr9 | NKX2-5 target | −0.0669 | −2.17 | 0.0341 | 0.545 |
| ALDOA[d] | chr16 | NKX2-5 target | −0.0646 | −2.17 | 0.0341 | 0.545 |
| TCAP[a] | chr17 | NKX2-5 target | −0.0178 | −0.282 | 0.779 | 1.00 |
| TOM1L2 | chr17 | NKX2-5 target | −0.0771 | −1.75 | 0.0849 | 1.00 |

Proteomics differential abundance results in human atrial appendage tissue for prevalent AF. Two-sided t-tests were calculated as part of a multiple linear regression model including AF-related covariates sex, age, BMI, diabetes, systolic blood pressure, hypertension medication, myocardial infarction and smoking status (see methods differential protein analysis, $N = 78$, df = 66). The Benjamini-Hochberg procedure was used to asses FDR and account for multiple comparisons.
AF atrial fibrillation, BMI body mass index, QTL quantitative trait loci, FDR false discovery rate.
[a]Mutation known to affect cardiovascular phenotypes.
[b]Mutation known to affect arrhythmias.
[c]Differential expression or functional impairment for cardiovascular phenotypes.
[d]Differential expression or functional impairment for arrhythmias.

improve our understanding of how genetic factors are related to intermediate molecular phenotypes in AF.

We found widespread genetic effects associated with the expression of nearby genes on transcript and protein level. Our integrated cis-eQTL and pQTL analysis allowed the distinction between functional regulatory mechanisms with consequences for mRNA and protein levels. For example, we found many genetic variants exclusively affecting mRNA or protein abundances contributing to a modest overlap between both molecular levels using stringent statistical criteria. In this regard, we and others found that proteome-specific pQTLs are enriched in the coding sequence[14], where post-transcriptional regulatory elements might be affected by sequence variants, which may at least partially explain the divergence between eQTLs and pQTLs. Compared to other studies, a similar extent of co-regulation between mRNA and protein levels was previously documented by comparing cis-pQTLs in human plasma to cis-eQTLs in GTEx tissue[15] (see Supplementary Note 1, Supplementary Table 2). We assume that the use of less stringent significance cutoffs or multiple testing correction as well as more sensitive measurement techniques might achieve a higher overlap as observed by Battle and colleagues for cell-type-specific transcriptomics and proteomics in the same lymphoblastoid cell lines[14]. In line with prior studies, we observed large differences in transcript and protein expression as well as their regulation[14,15,21], emphasizing the necessity and benefit of taking multiple molecular entities into account to investigate genotype–phenotype relationships.

To extend the cis-QTL analysis, we assessed trans-associations by applying a candidate selection strategy based on the correlation of gene expression with a PRS, a concept termed eQTS[12]. PRS accumulate small genetic effects at many individual genome-wide loci. In the theoretical omnigenic model[11] it has been suggested that these loci are linked to the phenotype by weak trans-effects on gene expression, which accumulate in so called core genes. It has been shown that this accumulation of trans-effects would lead to strong eQTS associations for core genes[12]. Here we used eQTS and pQTS in combination with gene set enrichment analysis to identify core gene pathways and putative core genes for AF. As core genes are postulated to have trans-associations with AF GWAS SNPs, we subsequently performed a targeted trans-QTL analysis. This strategy allowed the investigation of tissue-specific trans-acting genetic mechanisms in AF using a relatively small clinical dataset by reducing the multiple testing burden; however, it is limited to genes with functional annotations available. Here we used gene sets from gene ontology, to avoid introducing a bias towards already known disease gene set definitions, as for instance contained in KEGG. The PRS-based gene set enrichment approach revealed cardiac-specific pathways associated with the genetic susceptibility for AF, which are similar to results identified by Wang and colleagues[40]. We identified different pathways on transcriptome and proteome level, which is probably not only caused by biological but also technological reasons like protein coverage. On transcriptome level, the majority of the identified pathways were involved in contractile function and metabolism. In general, these mechanisms have been reported by clinical and experimental studies to play a major role in the pathology of AF[41–43]. As expected, also the putative core genes identified by trans-eQTLs and trans-pQTLs were involved in those mechanisms. In addition, all identified transcripts and some of the proteins have been described in the context of arrhythmias or cardiovascular disease[29–34,44–50] (Supplementary Table 12, differential expression results in Supplementary Tables 13-14). As observed for the cis-QTL analysis, the trans-analysis revealed similar differences between transcriptomics and proteomics level. Interestingly, none of the trans-pQTLs had a significant association on expression level (see Supplementary Table 17). Differences between trans-eQTLs and pQTLs for the same gene have previously not been discussed

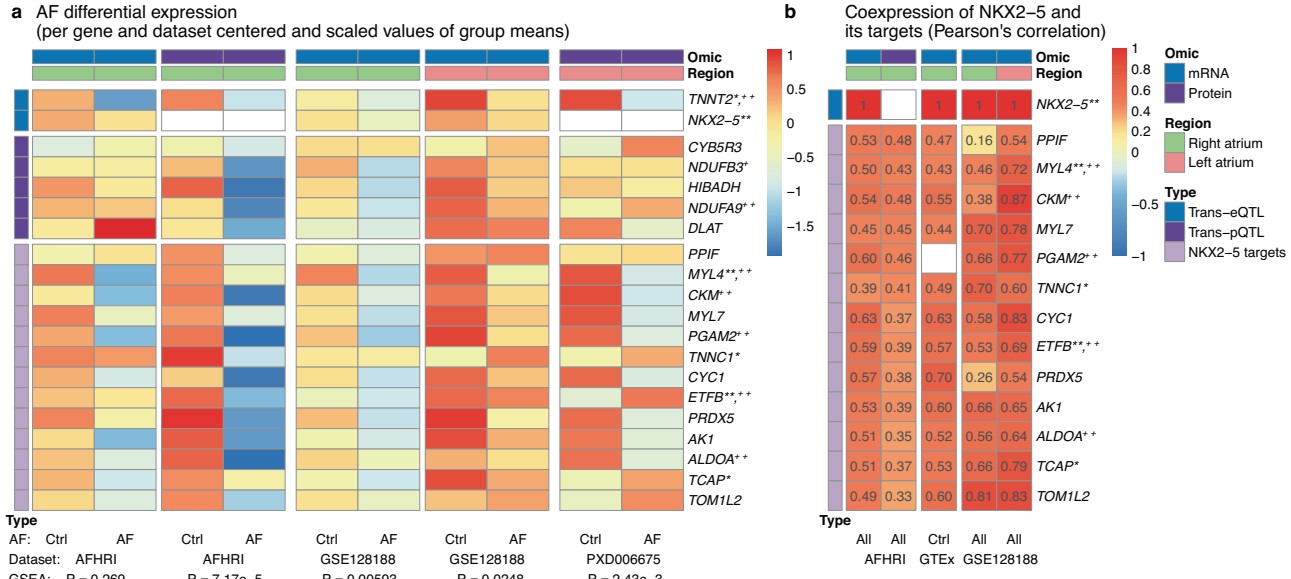

**Fig. 6 Replication of the core gene candidate AF association and NKX2-5 target coexpression in independent datasets.** Published proteomics data (PXD006675) as well as RNA-seq data (GSE128188, GTEx) generated from human atrial tissue samples were used for replication. **a** Centered and scaled values of the mean mRNA or protein expression in AF ctrls and cases, with stronger effects on protein level. GSEA p-values quantify the negative association of NKX2-5 targets with respect to AF. Sample sizes per column: 69 controls, 14 prevalent AF cases, 69 controls, 14 prevalent AF cases (AFHRI, all right atrial appendage); five controls, five AF cases (GSE128188, both right atrial appendage); five controls, five AF cases (GSE128188, both left atrial appendage); three controls, three AF cases (PXD006675, both left atrium). A quantitative description of the qualitative results presented in the heatmap can be found in Supplementary Table 13-14 and Table 3. **b** Coexpression of NKX2-5 with the 13 identified NKX2-5 transcription factor targets (Pearson's correlation). Quantified is the correlation between NKX2-5 and its targets on mRNA level for mRNA datasets and the correlation between the NKX2-5 transcript expression with the target protein concentrations for the AFHRI proteomics (NKX2-5 not quantified in proteomics). Sample sizes used for the computation of correlations: 102 AFHRI mRNA, 96 AFHRI protein, 372 GTEx, 10 GSE128188 right, and 10 left atrial appendage samples. AF atrial fibrillation, Ctrl control i.e. individuals in sinus rhythm, GSEA gene set enrichment analysis *Mutation known to affect cardiovascular phenotypes, **Mutation known to affect arrhythmias, +Differential expression or functional impairment for cardiovascular phenotypes, ++Differential expression or functional impairment for arrhythmias. Source data are provided as a Source Data file.

in detail in the literature. Sun and colleagues analyzed overlapping *cis*-QTLs but not *trans*-QTLs[15]. Yao and colleagues analyzed overlaps of *cis*- and *trans*- eQTLs and pQTLs in plasma[16], however no overlaps were found in *trans*. Suhre and colleagues validated plasma pQTL findings in other proteomics datasets, but did not evaluate corresponding *trans*-eQTLs[13]. Possible reasons besides the small sample size might be the effect of other genetic variants, post-transcriptional regulation or environmental factors.

To investigate more complex molecular mechanisms underlying AF, we further analyzed the TF network of NKX2-5, since the TF has been described in the context of heart development and arrhythmias. A loss-of-function mutation in *NKX2-5* is associated with increased susceptibility to familial AF[32] demonstrating its causal role for AF. However, this result still does not provide mechanistic insights. A study of allele specific binding of NKX2-5 identified relevant targets and demonstrated that variants that change NKX2-5 binding at the promoters of target genes contribute to electrocardiographic phenotypes[38]. Hence this study established a mechanistic link of NKX2-5 to AF susceptibility at common GWAS variants, which alter *cis*-regulatory elements bound by NKX2-5 at the target genes. In this work, we link the expression of *NKX2-5* to an AF-associated GWAS variant in *trans*. Moreover, we investigated downstream effects by integrating the functional data[38] with our genotype and expression data. We identified a set of target genes, where the AF-associated variant did not alter the *cis*-regulatory elements but the activity of the *trans*-acting TF NKX2-5. Taken together, prior studies

demonstrate a causal link of NKX2-5 with AF and suggested that AF-associated variants alter *cis*-regulatory elements. Here we showed that in addition *trans*-acting mechanisms are important to link AF-associated variants to their downstream target genes. The high correlation between the estimated NKX2-5 TF activity and its mRNA levels implies that NKX2-5 modulates target gene expression. The fact that the *NKX2-5* eQTL is a key regulatory mechanism prompted us to validate the *trans*-QTL on protein level with western blot analyses. Furthermore, the TF activity that was estimated based on genome-wide transcriptomics data as well as independent tissue and cell-type-specific annotations, highly correlated with actual protein intensities. While measured mRNA and protein abundance show a stronger association with the rs9481842 genotype than the estimated TF activity, correlation between the TF activity and protein measurements was higher compared to the correlation between mRNA and protein. Consistent with other studies[37,38], our analysis suggests that NKX2-5 acts as a transcriptional activator for the majority of genes. However, the TF can also function as a transcriptional repressor of genes like *ISL1*[37,51]. We were able to detect strong effects of the *NKX2-5* transcript on various target transcript and protein levels. Most of the identified target genes are involved in contractile function or metabolism, two mechanisms highly linked to processes involved in AF. Our unique *trans*-QTL approach revealed direct disease-relevant associations between candidate core genes, NKX2-5 target genes and AF. These genes were identified based on molecular data in our cohort in combination with public AF annotations without using the actual cohort phenotypes. The

collective differential protein abundance of NKX2-5 targets in AF patients compared to controls therefore serves as an independent validation of the disease relevance, which we independently replicated in additional RNA-seq[36] and proteomics[30] datasets of AF patients and controls.

Overall, we successfully integrated multi-omics data and established a unique approach to investigate not only *cis-* but also *trans-*regulatory effects. This provided a platform to generate hypotheses on functional interactions underlying the genetic associations that can be further experimentally investigated.

We acknowledge some limitations that are attributed to common biological and technical factors. First, the use of human heart tissue came with several challenges including restricted sample sizes and heterogeneity of cellular composition. The small sample size affects the statistical power of QTL analyses and does not allow for assessing causality of molecular and physiological changes, for example by Mendelian randomization[52,53]. Changes in cell-type composition and structural remodeling have been described for the pathology of AF[54]. To take differences in the cellular composition into account, we used a fibroblast-score based on a fibroblast-specific gene signature to adjust expression levels in eQTL/pQTL analyses[55]. Furthermore, although AF prevalently originates from pulmonary vein ostia, the relevance and usefulness of right atrial appendage tissue was demonstrated by prior studies which identified various AF disease mechanisms and candidate genes[3,56,57]. Therefore, we believe that the extracted tissue samples are well suited and the best proxy for atrial impairment for our analysis. Another strength is that tissue samples were explanted during cardiac surgery and not post-mortem as in comparable datasets, which can affect various pathways e.g. metabolism. Second, expression data were generated using microarrays, however, to date, more information can be generated by RNA-seq. Third, human cardiac muscle tissue is dominated by mitochondrial and sarcomere proteins[58], which affects the detection of less abundant proteins such as TFs (e.g. NKX2-5). Therefore, missing TF coverage was not due to data quality but biological and technological restrictions. In addition, only limited functional genomic annotations specific for atrial tissue are currently available including TF-, miRNA- and RBP binding sites. Therefore, we integrated multiple sources to render functional annotations as reliable and accurate as possible. Replication in an independent dataset was not feasible, since this was the first study investigating pQTLs in atrial tissue. Due to this restriction, literature research was carried out to validate the relevance of the identified AF core genes and NKX2-5 targets in the context of arrhythmia (Supplementary Table 12, differential expression results in Supplementary Table 13-14). Furthermore, we validated the association of target genes with AF by analyzing the differential expression on protein level between AF and controls. The consistent downregulation of the 13 NKX2-5 targets in AF cases compared to controls was replicated in two independent datasets including RNA-seq data from left and right atrial samples as well as proteomics from left atrial tissue. Additionally, coexpression of *NKX2-5* mRNA with transcript levels of the 13 targets was replicated in two independent datasets. Taken together this evidence suggests the identified genes as strong candidates for follow-up analyses. Yet, for clinical translation, further validations and molecular characterization are required.

In this study we suggest an integrative analysis of genomics, transcriptomics and proteomics data of human atrial tissue to identify genome-wide genetic effects on intermediate molecular phenotypes in the context of AF. Our multi-omics approach permits the identification of shared and independent effects of *cis-*acting variants on transcript expression and protein abundance. Furthermore, we proposed a PRS-guided analysis strategy to successfully investigate complex genetic networks even with a limited sample size. By providing these unique tissue-specific omics results as a publicly accessible database in an interactive browser, we hope to extend the availability of valuable resources for hypothesis generation, experimental design and target prioritization.

## Methods

Analyses were performed using R 3.4.1 and 3.6.3 (r-project.org). Genomics data in R was handled using the Bioconductor packages rtracklayer[59] 1.46.0 and GenomicRanges[60] 1.38.0.

**Patient cohort.** Patients were consecutively enrolled in the ongoing observational cohort study AFHRI-B (Atrial fibrillation in high risk individuals-biopsy) independent of AF disease status. Participants were older than 18 years of age and were scheduled to undergo open heart coronary artery bypass surgery. Any patients with other bypass surgeries or additional procedures, e.g. valve surgery, were excluded. For the current analyses, $N = 118$ patients with multi-omics data were available. Omics measurements were performed in batches (one batch genotypes, two batches transcriptomics both with approximately equal distribution of cases and controls and one batch of proteomics) depending on the amount and quality of the material and resource availability. Thus, the number of individuals who entered analyses differed by omics type. Information on classical cardiovascular risk factors and potential confounders (age, sex, body mass index, systolic and diastolic blood pressure, hypertension, hypertension medication, diabetes, diabetes medication, history of myocardial infarction, smoking) was collected by questionnaire and from medical records. Non-valvular prevalent AF was the clinical diagnosis based on patient history and routine cardiology work-up, that was used as outcome in our analysis. Baseline blood samples were aliquoted and stored prior to surgery. Right atrial appendage tissue remnants were collected when the extracorporeal circulation was started and shock frozen immediately. Follow-up for AF and other cardiovascular disease outcomes was done by questionnaire, telephone interview and medical chart review. The observational cohort study was approved by the Ethikkommission Ärztekammer Hamburg (PV3982). The study was performed in compliance with the Declaration of Helsinki. The study enrollment and follow-up procedures were in accordance with institutional guidelines. All participants provided written informed consent. Sex stratification of the results was not possible due to the inherently small number of women in the study (Supplementary Table 18). Analyses were adjusted for sex where appropriate. Analyses were performed in all samples with respective omics data that passed appropriate quality control as stated in the preprocessing steps. This resulted in slightly different samples analysed in eQTL/eQTS, pQTL/pQTS and differential protein expression analysis. Baseline characteristics of the cohort stratified by analysis type can be found in the supplement (Supplementary Table 18).

**Genotypes.** The genotype data were generated using the Affymetrix GeneChips Genome-Wide Human SNP Array 6.0, with quality control (QC) at different levels. Using the Birdseed v2 algorithm, PLINK 1.9 and standard quality control procedures[61], 749,272 SNP were identified in 83 blood samples with a MAF > 0.01, HWE exact test $P > 1 \times 10^{-6}$ and a call rate > 98%. Genotypes were further imputed with IMPUTE2[62] 2.3.2 based on the 1000 genomes Phase 3 genotypes[63,64] [https://www.internationalgenome.org/] (per SNP: confident genotype calls with genotype probability > 95%, percentage of confident genotype calls across samples > 95%) and included only variants with HWE $P > 1 \times 10^{-4}$ resulting in 5,050,128 SNPs for 83 individuals. All samples were of central European ancestry, no close relatedness (max. IBD = 0.04 ≪ 0.19) or population substructure could be detected.

For QTL analyses, outliers in the expression data which coincide with rare genotypes can lead to false positive findings. For SNPs with less than three individuals with the homozygous-minor-allele genotype, all samples with homozygous-minor-allele genotype were therefore recoded to heterozygous genotype.

**PRS for AF.** The polygenic risk score was calculated based on the LDpred omnigenic score for AF published by Khera et al.[6]. To account for a realistic representation of risk score values across the general population, we calculated risk score values together with unrelated 1000 genomes[63,64] CEU individuals. Phase 3 1000 genomes genotypes were filtered for variants in the risk score and merged with our AFHRI genotypes, resulting in SNP data for 6,730,540 variants out of 6,730,541 in the score. The PRS per individual was computed using the Plink 1.9 score function, imputing missing variants based on the frequency of the risk allele. From this, percentiles across all 490 individuals (1000 genomes: 407, AFHRI cohort: 83) were further used as PRS values for further analysis.

**mRNA.** The mRNA data were generated from human heart atrial appendage tissue samples obtained during heart bypass surgery. They were frozen in liquid nitrogen and pulverized for further analysis. RNA isolation was performed with subsequent assessment of the RNA integrity number (RIN) for quality determination of the samples. HuGene 2.0 ST Arrays were used with the Affymetrix® GeneChip WT

Plus Reagent Kit, the Affymetrix GeneChip 3000 scanner and the Affymetrix Genechip Command Console 4.0.0.1567. The R Bioconductor package oligo[65] 1.50.0 was used to create expression sets, perform the background correction and quantile-normalization per sample, as well as log-transform the data. Left atrial appendage tissues and samples with a RIN-score smaller than 6 were excluded, in case of replicates the one with the highest RIN-score was used. The mean of multiple transcript clusters annotated to the same gene symbol was used to derive gene level expression values for 26,376 genes in 102 samples.

**Protein**. To measure the protein concentrations of 97 right atrial appendage samples, the tissues were homogenized using a micro dismembrator (Braun, Melsungen, Germany) at 2600 rpm for 2 min in 100 μl of 8M urea/2M thiourea (UT). Then homogenates were resuspended in 300 μl of UT. Nucleic acid fragmentation was gained by sonication on ice three times for 5 s each with nine cycles at 80% energy using a Sonoplus (Bandelin, Berlin, Germany). The homogenates were centrifuged at $16,000 \times g$ for 1 h at 4 °C. After that, protein concentration was determined by Bradford with BSA as standard (SE). Three micrograms protein were reduced and alkylated and digested with LysC (1:100) for 3 h followed by tryptic digestion overnight both at 37 °C. Subsequently peptide solutions were desalted on C18 material (μ ZipTip). Finally mass spectrometry analysis was performed on a LC-ESI-MS/MS machine (LTQ Orbitrap Velos). One sample was excluded due to irregularities in the chromatographic pattern. The Rosetta Elucidator 3.3 workflow was used to extract feature intensity and derive protein intensities by summing of all isotope groups with the same peptide annotation for all peptides annotated to one protein (further parameters: Uniprot_-Sprot_human_rel. 2016_05: static modification: carbamidomethylation at Cys, variable modification: oxidation at methionine, 2 missed cleavages, fully tryptic, filtered for peptides with FDR < 0.05 corr. to Peptide Teller probability > 0.94 and shared peptides were excluded). Intensities for 1419 proteins with one or more peptides (877 with 2 or more peptides) were quantified for 96 samples, median-normalized and log10-transformed. The original protein concentration was determined as an important technical covariate and therefore used in further analyses.

For cis-QTL computations, matched genotypes and proteomics data were available for 1337 proteins in 75 samples including 62 missing values (0.06% of all values) that were imputed using the KNN-method implemented in the R bioconductor package impute 1.50.1.

**Protein analysis using western blot**. Human atrial tissue samples (15 mg each) were pulverized in liquid nitrogen and lyzed with M-PER Mammalian Protein Extraction Reagent (Thermo Scientific) supplemented with protease inhibitor. Protein concentrations were measured using a BCA assay (Thermo Scientific). The same amount of protein for each sample was heated at 95 °C for 10 min in 1× Laemmli. Proteins were separated on a 10% SDS-PAGE gel and transferred to nitrocellulose membranes. Membranes were blocked with 5% skim milk in TBS-T for 1 hour. Staining with the primary antibody was performed overnight at 4 °C, and secondary antibody staining for 1 h at room temperature. The following primary antibodies were used: NKX2-5 (ab205263, 1:1000), alpha-actinin (CST #3134, 1:1000), GAPDH (CST #3683, 1:2000). The following HRP-conjugated secondary antibody was used: goat anti-rabbit IgG (PI-1000-1, 1:10,000). The antibodies were visualized with enhanced chemiluminescence (ECL) detection reagent (Bio-Rad #1705060) or the SuperSignal West Pico PLUS chemiluminescent substrate (Thermo Scientific #34579). The membranes were reprobed with GAPDH antibody after incubation with stripping buffer (Thermo Scientific #46430) for 4 min, washing and blocking with 5% skim milk in TBS-T. Antibody detection was performed with a chemiluminescence imaging system (FUSION Solo S). Blot analyses were achieved with the Image Lab software (Bio-Rad 6.1).

**Protein-per-mRNA ratios**. mRNA and protein measurements were already per-sample quantile-normalized and log-transformed. Both mRNA and protein measurements were additionally quantile-normalized per gene and the ratio was computed as the difference between protein and transcript values.

**Residuals**. Per-sample quantile-normalized, log-transformed mRNA and protein values were used to compute residuals. mRNA residuals were derived as the residuals from a linear model explaining mRNA by protein levels, i.e. mRNA $\sim \beta_0 + \beta_1 \cdot$ protein $+ \varepsilon$. Protein residuals were derived as the residuals from a linear model explaining protein by mRNA levels, i.e. protein $\sim \beta_0 + \beta_1 \cdot$ mRNA $+ \varepsilon$. Covariates were used for further analyses but not for the calculation of residuals.

**Correction for cell-type composition—fibroblast-score**. Tissue samples consist of different cell-type compositions. Samples with more fibroblasts probably contain less cardiomyocytes, one of the functionally most relevant cell-types in primary atrial appendage tissue. We used a fibroblast-score based on the sum of expression values of genes upregulated in fibroblasts compared to cardiomyocytes in rats[55]: ELN, FGF10, FOSB, FCRL2, SCN7A, ARHGAP20, CILP, FRAS1, DCDC2, NRG1, AFAP1L2, ITGBL1, NOV, CLEC3B. Cardiomyocyte specific gene signatures were avoided to prevent interfering effects due to structural remodeling common in AF.

**Genome annotations**. Ensembl BioMart[66] GRCh37.p13 hg19 annotations were used as genome annotations.

**GWAS catalog**. GWAS annotations were based on the GWAS catalog[67] ([https://www.ebi.ac.uk/gwas/], 2019-11-26). We looked at the traits annotated to cardiovascular measurements (EFO_0004298) and cardiovascular disease (EFO_0000319), further referred to as cardiovascular traits. We also distinguished the subcategories arrhythmias, i.e. all traits annotated to atrial fibrillation, cardiac arrhythmia, sudden cardiac arrest, supraventricular ectopy, early cardiac repolarization measurement, heart rate, heart rate variability measurement, P wave duration, P wave terminal force measurement, PR interval, PR segment, QRS amplitude, QRS complex, QRS duration, QT interval, R wave amplitude, resting heart rate, RR interval, S wave amplitude or T wave amplitude, i.e. all traits annotated to Atrial fibrillation or QT interval based on the EFO-mapping (https://www.ebi.ac.uk/gwas/api/search/downloads/trait_mappings, 2019-11-26).

**VEP**. Ensembl Variant Effect Predictions[68] were downloaded from the Ensembl Biomart GRCh37.p13 based on SNP rs-IDs. The label Missense was used to summarize all possible missense consequences of the variant (gained stop codon, a frameshift/amino-acid altering/protein-altering variant, a lost start/stop codon, an inframe insertion/deletion).

**Chromatin states**. Roadmap Epigenomics ChromHMM 15 state model coremarks for human heart right atrial appendage[69] (E104_15_coreMarks_dense.bed) were used to annotate tissue-specific chromatin states.

**Promoter-capture HiC**. Promoter-capture HiC data from human iPSC-derived cardiomyocytes[70] E-MTAB-6014 (capt-CM-replicated-interactions-1kb.bedpe) was used to annotate linked promoter regions.

**Binding sites**. TF BS were based on ChIP-seq data from the ReMap TF database[71] (ReMap2018 v1.2) filtered for highly expressed genes (log(TPM + 1) ≥ 1) in GTEx atrial appendage tissue. Additionally, NKX2-5 binding sites from human iPSC-derived cardiomyocytes[38] GSE133833 were used. All TF BS were filtered for a minimal overlap of 25 bp with open chromatin regions, i.e. chromatin states 1_TssA, 2_TssAFlnk, 10_TssBiv, 6_EnhG, 7_Enh, 11_BivFlnk or 12_EnhBiv.

Fine mapping for functional NKX2-5 BS was done integrating promoter, promoter-capture HiC, chromatin states and NKX2-5 ChIP-seq data. Promoter regions were annotated based on Gencode[72] v31lift37 basic and long non-coding RNA transcript start annotations as well as regions linked to those by promoter-capture HiC. ChIP-seq binding sites were further overlapped with those promoter regions and filtered for open chromatin regions (details see provided analysis code).miRNA BS were based on TargetScan 7.2 default predictions for conserved target sites of conserved miRNA families[73]. RBP BS were derived based on eCLIP data from HepG2 and K562 cell lines provided by the ENCODE Project Consortium[74,75] (ENCODE, Supplementary Table 19). Peak calling was done using the ENCODE uniform processing pipeline, peaks in the bed-files were further filtered for an enrichment > log2(1), a Fisher P-value > −log10(0.05) and overlapping peaks were then merged (details see provided analysis code).

**Cis-QTL covariates including PEER factors**. Tissue-specific expression analyses remain challenging due to a large number of confounders. PEER factors[17] are widely used to account for known and unknown factors in the context of cis-QTL analyses[18,19] and were computed using the R package PEER 1.0 . One to 30 PEER factors without additional covariates, with fibroblast-score only, with the first three genotype principle components, with age, sex, BMI, disease status, fibroblast-score and with age, sex, BMI, disease status and fibroblast-score and three genotype principle components were used as covariates in the QTL analysis (Supplementary Fig. 1).

**Cis-QTL computation**. QTLs were calculated using the R package MatrixEQTL[76] 2.2. A cis-range of $1 \times 10^6$ bp and a linear, additive model for genotype effect was used. Expression quantitative trait loci (eQTL), protein quantitative trait loci (pQTL), expression residual quantitative trait loci (res eQTL), protein residual quantitative trait loci (res pQTL) and protein-per-mRNA ratio quantitative trait loci (ratioQTL) analyses were performed for per-sample quantile-normalized as well as additional per-gene quantile-normalized expression values, each for the different sets of covariates as described above.Normalization and covariate sets were optimized for the highest number of QTL genes (i.e. genes, with at least one significant QTL, as previously established by Lappalainen and colleagues[18] and the GTEx consortium[19]), detected based on a FDR < 0.05 (Benjamini-Hochberg procedure, Supplementary Fig. 1). For the final analysis, QTLs were computed using per-sample and per-gene quantile-normalized expression values, using only PEER factors without additional covariates. Two-sided t-tests were performed with 12 PEER factors for eQTLs (N = 75, df = 61), 10 PEER factors for pQTLs (N = 75, df = 63), 8 PEER factors for res eQTLs (N = 66, df = 56), 12 PEER factors for res pQTLs (N = 66, df = 52) and 9 PEER factors and the fibroblast-score for ratioQTLs (N = 66, df = 54).

To investigate independent QTLs for the same gene, LD clumping was performed using the Plink 1.9 clump function with parameters –clump-r2 0.5, –clump-kb 250, FDR cutoffs –clump-p1 0.05 and –clump-p2 0.8 as well as $P$ value cutoffs –clump-p1 $1 \times 10^{-5}$ and –clump-p2 0.05 for each gene and QTL type.

**Definition of functional QTL categories.** Shared eQTL/pQTL were defined as SNP-gene pairs with a significant eQTL (FDR < 0.05), pQTL (FDR < 0.05) and no res eQTL (FDR < 0.05) or res pQTL (FDR < 0.05), i.e. genetic regulation is observable on transcriptomics and proteomics level and variation corresponding to the SNP influence in one omic level can be explained and therefore removed by the variation in the other omic level.

Independent eQTLs were defined as SNP-gene pairs with a significant eQTL (FDR < 0.05) and res eQTL (FDR < 0.05) but no pQTL (FDR < 0.05) and no res pQTL (FDR < 0.05), i.e. regulation of SNP is only affecting transcript levels, not proteins. Also, the res eQTL disappears, if the SNP influences protein levels too much, for example a pQTL that barely missed the significance threshold.

Independent pQTLs were defined as SNP-gene pairs with a significant pQTL (FDR < 0.05) and res pQTL (FDR < 0.05) but no eQTL (FDR < 0.05) and no res eQTL (FDR < 0.05), i.e. regulation of SNP is only affecting protein levels, not transcripts, i.e. by post-transcriptional regulation.

**Colocalization analysis.** Colocalization analysis was performed using the coloc.abf() function from the R package coloc[22] 3.2-1. Posterior probabilities > 0.5 were used to classify colocalization. For independent eQTLs and independent pQTLs, we considered the sum of the posteriors for H1/H2 and H3 representing either only one independent QTL or independent eQTL and pQTL.

**Enrichment of functional elements.** Similar as described by Battle and colleagues[14], annotations of the top 5 QTL hits per gene were compared to a background distribution (100 background SNPs per QTL SNP) matched for MAF (difference ≤0.05) and distance to TSS (difference ≤ 1000 bp). Top QTL SNPs per gene were ranked according to the FDR of pQTLs for shared eQTLs/pQTLs, res eQTLs for independent eQTLs and res pQTLs for independent pQTLs. Odds ratios were computed using Fisher's exact test (two-sided) on the QTL-by-annotation contingency tables.

**GWAS overlap and enrichments.** To determine the overlap between GWAS hits and cis-QTLs, we first annotated all GWAS hits for cardiovascular traits and RA in the GWAS catalog[67] with proxies in high linkage-disequilibrium using SNiPA[77] ([https://snipa.helmholtz-muenchen.de/], EUR population, $R^2 > 0.8$) as well as significant QTLs ($P < 1 \times 10^{-5}$). For each of the original GWAS SNPs, the corresponding proxy-gene pair with the strongest QTL was selected to annotate this GWAS hit.

To assess a general enrichment of GWAS hits in QTLs, for all tested QTL SNPs we constructed the cross tables that a SNP has significant QTL ($P < 1 \times 10^{-5}$) versus was the SNP (or $R^2 > 0.8$ proxy) annotated in the GWAS catalog. These tables were evaluated for eQTLs and pQTLs for each of the groups cardiovascular traits, arrhythmias, AF and RA using Fisher's exact test (two-sided).

**PRS correlations/eQTS/pQTS rankings.** Transcriptomics and proteomics ranking based on PRS correlations were evaluated using linear regression models with additional covariates age, sex, BMI, systolic blood pressure (sysBP), C-reactive protein (CRP) and N-terminal prohormone of brain natriuretic peptide (NT-proBNP) as well as the lead SNP for each independent cis-QTL loci (based on LD clumps, FDR < 0.05) to correct for potential cis-effects included in the PRS. We assume that genetic-centered analyses are less susceptible to confounding than differential expression analyses. While we still included the most relevant cardiovascular risk factors and technical covariates RIN-score or protein concentration, we reduced the number of highly correlated covariates (e.g. blood pressure, hypertension and hypertension medication) and preferentially included continuous covariates to avoid overfitting/overcorrecting and to increase power. The following models were evaluated:

$$mRNA \sim \beta_0 + \beta_1 \cdot PRS + \beta_2 \cdot age + \beta_3 \cdot sex + \beta_4 \cdot BMI + \beta_5 \cdot sysBP + \beta_6 \cdot CRP$$
$$+ \beta_7 \cdot NT\text{-}proBNP + \beta_8 \cdot RIN + \sum_i \beta_i SNP_i + \varepsilon$$

$$protein \sim \beta_0 + \beta_1 \cdot PRS + \beta_2 \cdot age + \beta_3 \cdot sex + \beta_4 \cdot BMI + \beta_5 \cdot sysBP + \beta_6 \cdot CRP$$
$$+ \beta_7 \cdot NT\text{-}proBNP + \beta_8 \cdot protein\ conc. + \sum_i \beta_i SNP_i + \varepsilon$$

We further used summary statistics ($T$ statistic) for $\beta_1$, equivalent to comparing the nested models with/without the PRS and derived corresponding two-sided $P$ values. Degrees of freedom for genes without significant cis-loci were 65 for mRNA ($N = 74$) and 64 for protein ($N = 73$). Different numbers of samples compared to cis-QTL analyses are due to missingness in the used covariates.

**Pathway enrichment analysis.** Gene set enrichment analysis (GSEA)[27] was performed using the Bioconductor R package fgsea[28] 1.8.0 and MSigDB v6.1 gene sets

for Gene Ontology biological processes (c5.bp.v6.1.symbols.gmt.txt[27,78,79]). To avoid bias towards gene sets specific to human disease as e.g. KEGG pathways, Gene Ontology gene sets, which are not linked to a disease a priori were favoured. The GSEA method was selected to further identify the leading edge genes, which represent the drivers of the enrichment. Enrichments were calculated with 100,000 permutations on eQTS $T$-values (considering gene sets with minimal 15 and maximal 500 transcripts) and pQTS $T$-values (considering gene sets with minimal 5 and maximal 500 proteins).

**SNP and gene candidate selection for trans-analyses.** To reduce the multiple testing burden, trans-analyses were only performed on AF GWAS SNPs and candidate genes derived from the gene set enrichment analysis. We selected all SNPs with MAF ≥ 0.1 that were annotated with atrial fibrillation in the GWAS catalog[67] or the best proxy if the annotated SNP was not measured in our dataset. We further evaluated SNPs in high LD using SNiPA[77] ($R^2 > 0.5$) and took only the SNP with the highest $P$-value in a recent GWAS[3], resulting in 108 independent loci.

We performed power analysis for the ability to detect strong trans-eQTL effects with our fixed sample size ($N = 74$ for eQTLs). The trans-eQTL effect size was set to 21.8% of variance explained, which is the strongest trans-eQTL found in eQTLGen[80]. In particular we evaluated how many genes can be tested in a targeted trans-eQTL analysis of all LD pruned AF loci ($N = 108$ SNPs) to still have a power of at least 50% at a Bonferroni adjusted significance level of 5%, based on power calculations for the $F$ test (Supplementary Fig. 14). We found that 23 genes could be tested (Supplementary Fig. 14). Thus we designed our candidate selection strategy to identify the most promising 23 candidates.

Leading edge genes[27] defined by GSEA on the eQTS/pQTS associations were considered drivers of the enrichments of gene sets. A gene set was considered significantly enriched with a FDR < 0.05 (Benjamini-Hochberg procedure). This resulted in 1261 genes for 81 gene sets on transcript and 152 genes for one gene set on protein level.

Due to the hierarchical structure of the GO biological processes, we favoured genes that were driving the enrichment of multiple gene sets, i.e. also contained in smaller, more specialized child terms. For that reason, we selected all leading edge genes as trans-QTL gene candidates that appeared in the transcript leading edge set of 14 or more gene sets, reducing the 1261 to 23 genes (as based on the power analysis suggesting ≤23 genes).

Although protein candidates were much more abundant than 23 genes, because of only one significant gene set we could not apply the same selection strategy resulting in no further preselection.

**Trans-QTL computations.** Trans-QTLs were calculated using the R package MatrixEQTL[76]. An additive linear model was evaluated for 108 SNPs for AF and 23 transcripts as well as 152 proteins. Additional covariates age, sex, BMI, sysBP, CRP, NT-proBNP, the fibroblast-score and RIN-score/protein concentration for transcripts/proteins were used similar to the QTS analyses and resulted in two-sided $t$-test with 74 samples (df = 64) for transcripts and 73 samples (df = 63) for proteins. Different numbers of samples compared to cis-QTL analyses are due to missingness in the used covariates. In contrast to the cis-QTL analyses, no PEER factors were used as has been previously suggested for trans-analyses[18].

**NKX2-5 trans-pQTL evaluation.** The rs9481842-NKX2-5 trans-pQTL was evaluated using a linear model with additive genotype effect and covariates age, BMI, sysBP, CRP, NT-proBNP and the fibroblast-score to explain logarithmized NKX2-5 protein intensities normalized to alpha-actinin ($N = 29$, df = 21). Compared to the original trans-QTL computations, sex was dropped as a covariate since only one female sample was present.

**NKX2-5 target definition.** We were interested in investigating the link between a GWAS hit to target genes through a trans-eQTL-regulated TF, that was not measured on proteomics level. We therefore investigated the effect of the SNP as well as the TF transcript on target genes on transcriptomics and proteomics level. A graphical summary of the target definition procedure is shown in Supplementary Fig. 13. To prioritize target genes with most evidence of an association with the QTL SNP which is mediated by the TF, we aimed to establish the following properties:

(a) The target has a NKX2-5 binding site (ChIP-seq) overlapping with an open chromatin state in the promoter or an promoter interacting region (HiC). To establish that the TF mediates the effect of the QTL SNP on the target gene transcription we further need to show that:

(b) The transcript expression of the target gene is associated with the SNP genotype and

(c) the association between the SNP and the target transcript expression disappears when adjusting for TF expression levels.
Finally, the most relevant endpoint, i.e. target protein abundance, should be mediated by the TF.

(d)  The protein expression of the target is significantly and positively correlated with the transcript expression of the TF.

We do so by evaluating the following regression models (two-sided $t$-tests):

(1)  Association of GWAS SNP with target transcript (*trans*-eQTL, $N = 67$, df = 63): target transcript $\sim \beta_0 + \beta_1 \cdot \text{SNP} + \beta_2 \cdot \text{fibroblast-score} + \beta_3 \cdot \text{RIN} + \varepsilon$

(2)  Independent effects of the SNP on target transcript, that are not mediated by the TF transcript ($N = 67$, df = 62): target transcript $\sim \beta_0 + \beta_1 \cdot \text{SNP} + \beta_2 \cdot$ TF transcript $+ \beta_3 \cdot \text{fibroblast-score} + \beta_4 \cdot \text{RIN} + \varepsilon$

(3)  Association of target protein with TF transcript ($N = 79$, df = 75): target protein $\sim \beta_0 + \beta_1 \cdot \text{TF transcript} + \beta_2 \cdot \text{fibroblast-score} + \beta_3 \cdot \text{protein conc.} + \varepsilon$

(4)  Additionally, we quantify the corresponding *trans*-pQTL for Supplementary Table 16: Association of GWAS SNP with target protein (*trans*-pQTL, ($N = 66$, df = 62): target protein $\sim \beta_0 + \beta_1 \cdot \text{SNP} + \beta_2 \cdot \text{fibroblast-score} + \beta_3 \cdot \text{protein conc.} + \varepsilon$.

For direct binding a) we considered only genes with transcriptomics and proteomics measurements and at least one functional TF BS (see methods Binding sites: functional NKX2-5 BS).

To establish regulation by the SNP, we selected only genes with a significant association of the SNP with the target transcript (concordant effect to the TF expression) in regression model (1) $\beta_1 < 0$, $P(\beta_1) < 0.05$ to ensure (b) and additionally checked the vanishing effect when adding the TF transcript model, i.e. for (c) we assessed in regression model (2) $P(\beta_1) > 0.2$. For the remaining candidates, we performed FDR correction (Benjamini-Hochberg) on $P$-values of model (3). We finally identified all proteins that satisfy (d) by (3) FDR($P(\beta_1)$) < 0.05 and $T(\beta_1)) > 0$ as functional NKX2-5 targets.

**Partial correlations.** Partial correlations were computed using the R package ppcor[81] 1.1.

**Differential proteome analysis for AF.** Differential analysis of proteins was done by comparing protein expression of controls without AF to cases diagnosed with prevalent AF. Cases that developed only post-operative AF after surgery were excluded from the analysis. Since differential expression analyses with respect to AF might be more confounded by underlying conditions, more stringent adjustment was carried out by including covariates covs: age, sex, BMI, diabetes, sysBP, hypertension medication, myocardial infarction, smoking status, fibroblast-score and protein concentration. Summary statistics for $\beta_1(\text{AF})$ in the following linear model ($N = 78$) were used: protein $\sim \beta_0 + \beta_1 \cdot \text{AF} + \sum_{\text{cov} \in \text{covs}} \beta_{\text{cov}} \cdot \text{cov} + \varepsilon$.

**Replication in the GSE128188 dataset.** RNA-seq transcriptomics data of 20 atrial appendage tissue samples from males undergoing coronary artery bypass grafting and/or atrial/mitral valve repair or replacement was downloaded from the public repository GSE128188[36]. Norm factors were calculated to derive log-transformed TMM-based RPKMs using the functions calcNormFactors() and rpkm() from the bioconductor R package edgeR[82] 3.28.1. Full summary statistics for the AF differential expression analysis were supplied by the original authors after inquiry. GSEA was performed on the log fold changes while ranking significant genes before non-significant ones. For left and right atrial appendage samples each, all ten samples as well as the mean of cases and the mean of controls were scaled and centered per gene to produce relative values comparable across datasets.

**Replication in the PXD006675 dataset.** Mass spectrometry proteomics data of six left atrial tissue samples were downloaded from the public repository PXD006675[30]. Full summary statistics for the AF differential expression analysis were supplied by the original authors after inquiry. GSEA was performed on the log fold changes while ranking significant genes before non-significant ones. Proteomics data of triplicates for three AF cases and three controls were median-normalized per measurement and log-transformed. All 18 samples as well as the mean of the AF cases and the mean of the controls were scaled and centered per gene to produce relative values comparable across datasets.

**Reporting summary.** Further information on research design is available in the Nature Research Reporting Summary linked to this article.

## Data availability

The genotype, transcriptomics and proteomics data are available under restricted access, as they contain identifying participant information. Deposition in online repositories or controlled access repositories is not authorized by the patient's consent. Access to the complete data including phenotypes can be obtained by any qualified researchers as part of an academic or industry collaboration. Co-authorship on resulting publications is required only if authorship criteria are fulfilled according to the guidelines of good scientific practice of the Deutsche Forschungsgemeinschaft (10.5281/zenodo.3923602). Requests including a formal research proposal indicating the use of data and planned analyses should be addressed to Renate Schnabel (r.schnabel@uke.de) and will be processed within two weeks. Successful applications enable the unrestricted analysis of

the data in the context of cardiovascular disease. Mandated source data are provided with this paper. All results are available at http://qtldb.helmholtz-muenchen.de and in the Zenodo repository [https://doi.org/10.5281/zenodo.5080229][83]. For replication purposes, publicly available data were obtained from GTEx (RNA-seq and cis-eQTL results), GSE128188 and PXD006675. Additionally, the following annotations were used Ensembl BioMart GRCh37.p13: hg19 and Ensembl Variant Effect Predictions [http://feb2014.archive.ensembl.org/biomart/martview/], 1000 Genomes Phase 3 genotypes [https://www.internationalgenome.org/], GWAS catalog [https://www.ebi.ac.uk/gwas/] (2019-11-26), Roadmap Epigenomics E104_15_coreMarks_dense.bed [https://egg2.wustl.edu/roadmap/data/byFileType/chromhmmSegmentations/ChmmModels/coreMarks/jointModel/final/], E-MTAB-6014 capt-CM-replicated-interactions-1kb.bedpe [https://www.ebi.ac.uk/arrayexpress/experiments/E-MTAB-6014/], ReMap2018 v1.2 [http://pedagogix-tagc.univ-mrs.fr/remap/download/remap2018/hg19/MACS/remap2018_nr_macs2_hg19_v1_2.bed.gz], TargetScan 7.2 [http://www.targetscan.org/vert_72/vert_72_data_download/Predicted_Target_Locations.default_predictions.hg19.bed.zip], GSE133833 [https://www.ncbi.nlm.nih.gov/geo/query/acc.cgi?acc=GSE133833] and ENCODE eCLIP HepG2/K562 data [https://encodeproject.org]. Source data are provided with this paper.

## Code availability

The analysis code is available on github at https://github.com/heiniglab/symatrial [https://doi.org/10.5281/zenodo.5094276][84].

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

## Acknowledgements

This work results from a cooperation in the context with the eMed Junior systems medicine research alliance symAtrial, for which funding by the Federal Ministry of Education and Research was granted to M.H. (01ZX1408D), M.O.S. (01ZX1408B), R.S. and T.Z. (01ZX1408A). M.H. gratefully acknowledges funding by the Federal Ministry of Education and Research (BMBF, Germany) in the project eMed:confirm (01ZX1708G). R.S. has received funding from the European Research Council (ERC) under the European Union's Horizon 2020 research and innovation programme (grant agreement No 648131), from the European Union's Horizon 2020 research and innovation programme under the grant agreement No 847770 (AFFECT-EU) and German Center for Cardiovascular Research (DZHK e.V.) (81Z1710103). T.Z. received funding from the eMed Network (01ZX1708A), and from the German Center for Cardiovascular Research (DZHK e.V.) (81X2710170, partner site project; 81X2710105 Shared expertise project). E.H. was supported financially by the German Center for Cardiovascular Research (DZHK e.V.) (81X2400118 Shared expertise project). This project was also supported by the "Close the gap" funding of the UKE. Fabian Denbsky setup the prototype of *cis*-QTL processing pipeline during his Master's thesis. Tim Hartmann, Grit Höppner, Justus Stenzig, Jorge Duque Escobar and Teng Tong kindly assisted with laboratory support, Vishnu Dhople with mass spectrometric data acquisition and Toray Akcan supplied the processed eCLIP data to derive RBP binding sites. Johann Hawe supplied the setup hosting the analysis results. The researchers are indebted to the participants for their willingness to participate in the study and we are very grateful for the staff of the Clinical Cohorts Studies for their important contributions.

## Author contributions

T.Z., R.S. and M.H. conceived the research, supervised the study and obtained funding. J.K., U.V., E.H. and T.Z. performed experiments. R.S., C.S.B. and L.C. recruited the patient cohort and collected data. I.A. and M.H. developed computational methods and analyzed the data. B.G. and C.M. contributed to statistical results interpretation. M.O.S. analyzed genotype data. C.M. contributed to the analyses of transcriptomic data. I.A., J.K., T.Z. and M.H. wrote the manuscript with input from all authors. All authors made critical revisions of the manuscript and approved the final version of the manuscript.

## Funding

## Competing interests

The authors declare no competing interests.
