## [Peer Review File · Nature Communications]

Reviewers' Comments:

Reviewer #1:

Remarks to the Author:

Overall:

Heinig and colleagues present an interesting multiomic analysis of atrial tissue to glean new insights regarding atrial fibrillation. The analyses were thorough, comprehensive, and novel. I specifically enjoyed the analyses to prioritize core genes for atrial fibrillation, which the investigators are uniquely capable of doing with individual level data. I appreciate that the authors posted a preprint and their code.

Major:

1. In the calculation of the AF PRS, what proportion of variants were missing? What fraction of them were higher weight SNPs?
2. In coronary artery bypass surgery, atrial tissue that is typically sampled is atrial appendage. However, atrial fibrillation foci typically occur at the pulmonary vein ostia. Are the tissues extracted actually relevant to atrial fibrillation?
3. Disease status is unavailable in GTEx. However, the authors here have information regarding atrial fibrillation status. Instead of just using GTEx data for eQTL analyses, the paper would be enriched if they leveraged atrial fibrillation case/control data to identify atrial fibrillation genes.
4. Could the small proportion of eQTL/pQTL overlap be due to differences in heritability between transcription and translation? A correlation of effect sizes or directions may better compare eQTLs and pQTLs given likely such systemic differences.
5. In Fig 3, I would add one or two non-cardiovascular traits as a negative control to verify tissue specificity and that general enrichment of all disease-associated variants isn't simply being observed.
6. The authors assume that an AF PRS is a "proxy for an aggregation of AF-related trans effects across the whole genome." How are the authors excluding cis-acting variants associated with AF? With a full AF PRS, are they merely picking up strong cis effects?
7. Using summary-level data from sources such as GTEx, INTERVAL, SCALLOP, are the AF PRS effects on gene expression and protein concentrations described in the paper specific to atrial tissue or are such changes observed in other tissues?

Minor:

1. I would suggest changing "multiOMICS" to "multiomics" since "OMICS" is not an acronym.
2. cis and trans should be italicized.
3. Gene names should be italicized.
4. "Coronary bypass" should be "coronary artery bypass." Also, it's noted that "patients with surgery other than coronary bypass, e.g. valve surgery, were excluded." Many patients undergo simultaneous coronary artery bypass and valve surgery; I assume these individuals are excluded? If so, please clarify in the text.
5. Methods indicate that direct genotypes were filtered for HWE $P > 1e-6$, and imputed genotypes for HWE $P > 1e-4$. Shouldn't HWE $P < 1e-6$ and $< 1e-4$ be removed?
6. What's the rationale for recoding SNPs with < 3 homozygous for the minor allele to heterozygous?

Reviewer #2:

Remarks to the Author:

In this study, the authors propose an integrative analysis of genomics, transcriptomics and proteomics data of human atrial tissue to identify genome-wide genetic effects on intermediate molecular phenotypes in the context of AF.

Major comments

While this is a fascinating study, the reviewer questions the exact novelty of the study in terms of

mechanistic insight, and in particular, misses novel mechanistic data linking a specific gene / pathway to a specific mechanism contributing to AF. Such data would improve the manuscript considerably.

Minor comments

There is a lack of vital information in the Results section. This info can be found elsewhere in the manuscript, but its absence from the Results hinders reading. For example, the first paragraph of the Results misses important information concerning the tissue studied, its origin, number of samples etc. Similarly, for example, for Table 1. This needs to mention for example that this concerns atrial tissue and should mention the number of atrial samples studied.

It is unclear whether SNPs of Table 2 correspond to SNPs from AF GWAS (or their proxies).

Data, Figure 2b and 2c: can you please clarify that lack of detection of both eQTL and pQTL was not due to a statistical power issue in the corresponding dataset?

Reviewer #3:

Remarks to the Author:

The identification of over 100 loci associated with atrial fibrillation (AF) has enabled the generation of genome-wide polygenic risk scores (PRS) for AF risk. While rare monogenic forms of familial AF carry a high risk, the additive effects of many common genetic variants explain a much greater proportion of AF risk in the population. However, the precise mechanisms by which AF-associated single nucleotide polymorphisms (SNP) modulate the expression of multiple genes and pathways remains unclear especially as >95% of the polymorphisms occur in non-coding regions. Assessing tissue-specific cis-acting expression quantitative trait loci (eQTL) is a commonly used approach to decipher the SNP-gene relationship but this strategy explains only a small fraction of the identified AF risk loci. However, the contributions of trans-genetic variants, distant to the target gene, to the genetic architecture of complex polygenic traits are unknown. According to the theoretical omnigenic model, trans-effects may explain up to 70% of the heritability of a complex disease, like AF, in part through gene regulatory networks. The goals of this study were to establish methods to identify core AF genes targeted by gene regulatory networks, and to integrate genomic, transcriptomic, and proteomic data from human atrial tissue to better understand how the relationship between common genetic variants and molecular changes in AF and uncover genotype-phenotype relations. Assum and colleagues identified two trans eQTLs and 4 trans protein (p) QTLs for the GWAS SNP rs9481842 and AF. Collectively, this study provides a comprehensive multi-omics approach to identifying trans-acting networks in small datasets and provides a rich resource of atrial tissue-specific regulatory variants for transcript and protein levels for gene prioritization.

General Comments

Overall, Assum and colleagues propose a novel multi-omics approach that integrated genomics, transcriptomics, and proteomics data from human atrial tissue to identify the underlying molecular networks for AF and uncover genotype-phenotype relationships. The manuscript is well-written and the conclusions on the whole are supported by the data; the methods are highly innovative especially a novel targeted trans QTL approach based on PRS; the findings are intriguing and may potentially provide important insights into molecular mechanisms by which AF SNPs modulate genes and signaling pathways using both cis- and trans-genetic effects. The statistical analysis appears to be appropriate and sufficient detail is provided to reproduce the work. Addressing the following specific comments will greatly aid the reader in the interpretation of the findings and potential impact of the study.

Specific Comments

1. The additional 'PEER factors to adjust for known as well as unknown confounders' should be described in the main manuscript.
2. The replication rate of eQTLs in GTEx atrial appendage tissue was assessed. While the 'effect sizes

for the best eQTL ($P < 1 \times 10^{-5}$) per gene showed a correlation of 0.83 ($P = 3.6 \times 10^{-67}$) in GTEx, 66% replicated (GTEx $P < 1 \times 10^{-5}$) and 88% showed concordant allelic effects', what was the correlation for the least concordant eQTLs?

3. It is a little surprising that 'only 8.2% of significant SNP-gene associations are shared between mRNA and proteins (Figure 1, Suppl Figure S5, Suppl Table S2)'. The potential explanations for such a low shared rate between mRNA and proteins should be discussed especially with many studies using transcriptomic analyses to infer SNP-gene associations.

4. The investigators used the PRS as a proxy for an aggregation of AF-related trans-effects across the whole genome. However, many of the AF-associated SNPs incorporated into PRS regulate nearby genes, i.e., display cis-genetic effects. Thus, it is unclear why both cis- and trans-effects are included in the model.

5. The link between the core candidate genes and AF was established using either trans eQTLs or pQTLs for the AF GWAS hits. Surely it would be better to use the 21 genes with mRNA and protein concordance?

6. The number of right atrial tissue samples used for trans-QTL analyses is unclear and should be clearly stated in the manuscript. It is also important to discuss the limitations of using right atrial appendage tissue samples to investigate the association between common genetic variants and regulatory genes for AF.

7. The weak association between SNP rs8481842 and transcription factor (TF) is troubling as is the failure to detect NKX2-5 at the protein level in atrial tissue.

8. There are a number of limitations of the study that should be discussed fully: i) The small clinical dataset, and sample size and the heterogeneity of the underlying substrate for AF. ii) The GO gene sets were selected and failed to include calcium homeostasis, a key signaling pathway in the pathogenesis of AF. iii) Replication in an independent dataset was not performed.

9. Minor: i) Inconsistent use of abbreviations – too many to list. ii) The concept of different genetic regulatory patterns derived by multi-omics QTL integration is unclear and requires clarification. iii) Figure 2 should include P-values.

Reviewer #4:

Remarks to the Author:

I read with interest the article by Assum et al. on multi-omic analyses for atrial fibrillation. In this paper, the authors collect phenotypic, genotyping, transcriptomic and proteomic data from human heart atrial tissue from ~100 individuals undergoing coronary bypass surgery and jointly analyze the data to make inference regarding genes that are related to atrial fibrillation by acting on human atrial tissue. In doing so, they also create a resource of atrial eQTLs and pQTLs which they make publicly available. The study is generally well designed and executed although some methodological aspects need to be improved/clarified. In addition, power/sample size limitations dampen enthusiasm in the absence of independent validation.

Major concerns:

1. The authors undertake an interesting approach to identify "core genes" for atrial fibrillation by leveraging an eQTL approach followed by a targeted trans-QTL analysis. The design of the approach is reasonable but the trans-QTL associations remain underpowered (no association has an FDR < 0.05 even within this limited multiple testing burden). Consequently, it is necessary to provide evidence of independent samples.

2. Similarly, experimental validation of the NKX2-5 results would provide compelling support for the manuscript.

3. The authors should test for cross-mappability potentially explaining their trans-eQTL associations (see PMID: 30613398)

4. The authors selected the covariates to include in their QTL analyses methods based on what covariate combination results in the highest number of significant genes. While that method has been

used to select number of latent factors to include, it is not appropriate to not include known confounders (eg. age, sex, disease status) simply because they reduce associations. This could lead to false positives.

5. Similarly, genotype principal components should be included in all QTL analyses to correct for population substructure

6. A MAF threshold of 0.01 is too low for inclusion in this study of ~100 participants, they should use 0.05.

Minor concerns:

1. In addition to validation, the authors should make available in a supplemental table the top genes associated with the Afib PRS based on their QTS analyses with their corresponding association statistics.

2. Do the trans-QTL SNPs tested in Table 2 have a corresponding cis-QTL association? These should be reported, if so, and test whether these associations colocalize with the trans associations suggesting a shared causal signal.

3. Did the authors test for relatedness between individuals in their cohort? If so, how did they handle sample from related individuals?

4. The authors report that 8.2% of SNP-gene associations have both an eQTL and a pQTL. How many of those colocalize, suggesting a shared causal variant between the two signals?

5. The authors report 21 genes and 1083 pQTL variants that don't have a corresponding eQTL in their dataset. How many of those have a concordant eQTL in GTEx Atrial Appendage? Also, how many have a missense variant in their credible set that could explain the discrepancy?

6. Did the authors confirm that the Afib PRS is associated with atrial fibrillation in their cohort?

7. The authors need to provide further details on their method for prioritizing NKX2-5 targets for trans association testing. The process is a bit unclear from their methods as currently written. Also, what do the colors in Figure 5d represent? It currently reads as though these are z-scores but it's unclear how those were estimated for each genotype.

8. Figure 3c top panel: The pQTL estimate is annotated as having a p-value >0.05 according to the color but the 95% confidence intervals do not include 0. How were those calculated?

9. How was FDR calculated for Table 1? Is this gene level FDR or genome-wide?

10. There are some discrepancies in the numbers between the main text and tables/methods in several places: a) GWAS overlap and enrichment section: text mentions 17 AF-related eQTLs and 4 with pQTLs while Supp Table S4 lists 15 and 3, b) Protein section in the Methods reports measuring protein concentrations in 102 samples but intensities quantified for 96, c) cis QTL computation in Methods: There appear to be ~100 samples with RNA and protein quantification and 83 with genotype while cisQTLs are computed for 75 only – is that the overlapping set or were there additional exclusions?

Point by point response to reviewer comments

First of all, we would like to thank all reviewers for their comments and for having taken the time to review our work. We found the reviewers' comments very helpful and believe that the changes we have made based on the comments have strengthened our study.

Point by point response to reviewer comments	1
Reviewer #1 (Remarks to the Author):	2
Overall:	2
Major:	2
Minor:	10
Reviewer #2 (Remarks to the Author):	11
Major comments	11
Minor comments	12
Reviewer #3 (Remarks to the Author):	15
General Comments	15
Specific Comments	16
Reviewer #4 (Remarks to the Author):	26
Major concerns:	26
Minor concerns:	33

Reviewer #1 (Remarks to the Author):

Overall:

Heinig and colleagues present an interesting multiomic analysis of atrial tissue to glean new insights regarding atrial fibrillation. The analyses were thorough, comprehensive, and novel. I specifically enjoyed the analyses to prioritize core genes for atrial fibrillation, which the investigators are uniquely capable of doing with individual level data. I appreciate that the authors posted a preprint and their code.

We would like to thank the reviewer for highlighting that our study is thorough, comprehensive, and novel.

Major:

1. In the calculation of the AF PRS, what proportion of variants were missing? What fraction of them were higher weight SNPs?

For the 1000 Genomes individuals, almost all of the 6 730 540 SNPs with weights in the PRS were measured with an overall genotyping rate of 99.9%. For the AFHRI-B cohort, 33.5% had a genotyping rate per SNP lower than 80%. If we consider “higher weight SNPs” as the top quartile of SNPs ranked by their risk score weight, 57.5% of higher weight SNPs have a genotyping rate per SNP of less than 80.0%.

While these numbers appear to show a significant lack of information, we will show in the following analyses that our PRS still captures the relevant information needed.

For the 1000 genomes individuals, we were able to observe both the full PRS and the one restricted to only the non-missing variants in our smaller cohort and the comparison shows a very high correlation (**Figure R1a**). Also, the PRS used in the paper shows a stronger association with AF compared to other classical risk factors (**Figure R2, Supp Figure S8**).

Contribution of the missing SNPs to the PRS

High SNP weight, which is determined by large effect sizes of the original GWAS signal, is strongly correlated with a lower minor allele frequency which in turn influences missingness based on chip design and imputation accuracy.

For this paper, we used the plink score function to compute the risk score values per individual on our AFHRI cohort together with unrelated individuals from the 1000 genomes project by summing over the number of risk alleles multiplied by the SNP weight for each variant. In case of a missing genotype, the number of risk alleles is imputed based on the expected number with respect to the minor allele frequency estimated from all non-missing genotypes.

To quantify the amount of information contained in missing SNPs, we estimated the proportion of variance of the risk score captured by the different subsets of SNPs in the score evaluated for the 1000 genomes individuals only. Higher weight SNPs cover 79% of variance (R^2) compared to all SNPs, but even though less than half of higher weight SNPs were measured in the AFHRI cohort, all SNPs not missing in the AFHRI cohort cover 92% (R^2) of variance in the risk score.

Figure R1: Correlation of risk score values for different implementations of the genome-wide polygenic score for AF.

We further evaluated how strong the PRS calculated on the different subsets differed.

- For the 1000 genomes cohort, we compared the PRS on all SNPs in the score versus the subset that was non-missing in the AFHRI cohort (**Figure R1a**).
- For the AFHRI cohort, the same comparison was not possible due to the missing genotypes. Instead, we compared the score used in the paper to one computed on dosages for all genotypes irrespective of their imputation quality (**Figure R1b**).
- Finally, we compared the score calculated based on all imputed genotype dosages to a score including only the non-missing genotypes in the AFHRI cohort (**Figure R1c**).

Association of the PRS with phenotype

More importantly, for the three scores computed on the AFHRI cohort that were mentioned above, we also checked how well they discriminate between AF cases and controls (**Figure R2**).

Figure R2: Differential analysis for AF cases and controls for different implementations of the genome-wide polygenic risk score for AF.

The advantage of estimating the PRS together with the 1000 genomes individuals is that we can get a more realistic distribution of the PRS for the whole population by using the percentiles of the score instead of the original risk score values.

Logistic regression attributes the largest effect size to the genetic factors represented by the PRS compared to other common risk factors for AF, which we now also show in Suppl Figure S8 that has been added to the manuscript.

Figure S8: Genome-wide polygenic score adds relevant information in classifying atrial fibrillation disease status.

a: Percentiles of the atrial fibrillation polygenic risk score by disease status ($t=-2.0$, $P=0.026$, one-sided t-test).

b: Logistic regression results for common risk factors of AF. Significant and comparably strong effect for the PRS variable (estimate 4.7, $P=0.028$, two-sided t-test).

In the boxplots, the lower and upper hinges correspond to the first and third quartiles (the 25th and 75th percentiles). The median is denoted by the central line in the box. The upper/lower whisker extends from the hinge to the largest/smallest value no further than $1.5 \cdot IQR$ from the hinge.

AF, atrial fibrillation; PRS, genome-wide polygenic score; BMI, body mass index; BP, blood pressure; MI, myocardial infarction; CRP, C-reactive protein; NT-proBNP, N-terminal prohormone of brain natriuretic peptide.

2. In coronary artery bypass surgery, atrial tissue that is typically sampled is atrial appendage. However, atrial fibrillation foci typically occur at the pulmonary vein ostia. Are the tissues extracted actually relevant to atrial fibrillation?

To date, multiple studies have used atrial appendages to analyze and identify relevant disease-mechanisms or candidate genes on multiple molecular levels (Mayr et al., 2008; Roselli et al., 2018; Censi et al., 2010; Martin et al., 2015; Brundel et al., 2001). This is the tissue available for research since the pulmonary vein ostia cannot be used for safe biopsies. Furthermore, although AF in many cases originates from the pulmonary vein ostia, other mechanisms like electrical and structural remodeling of the atrial myocardium can represent substrates for AF initiation. Therefore, we believe that the extracted tissue samples are well suited and the best proxy for atrial impairment for our analysis. The use of this type of tissue renders our results comparable to prior research and will permit comparable research for validation. We have added the justification for using atrial appendage tissue in the discussion.

The relevance of right atrial appendage tissue for AF is further demonstrated by the clear enrichment of GWAS hits at significant eQTL and pQTL loci in our study. This enrichment was the strongest for AF GWAS hits compared to other GWAS studies for arrhythmias in general or other cardiovascular diseases (e.g. coronary artery disease). On that note, as proposed by the reviewer, we now also include the trait rheumatoid arthritis in the enrichment analysis as a negative control in the GWAS enrichment analysis.

3. Disease status is unavailable in GTEx. However, the authors here have information regarding atrial fibrillation status. Instead of just using GTEx data for eQTL analyses, the paper would be enriched if they leveraged atrial fibrillation case/control data to identify atrial fibrillation genes.

We would like to thank the reviewer for highlighting the distinct added value of our cohort, in which we have disease status available in addition to eQTL and pQTL data. While individual combinations of data (GWAS / PRS: SNP + AF phenotype, eQTL: SNP + expression, DEG: AF phenotype + expression) have been investigated previously, the ambition of our core gene approach is to identify gene-regulatory mechanisms involved in atrial fibrillation, bridging the gap between disease associated variants and functional genes. Our cohort puts us in the unique position to do so, as we have genetic, molecular and clinical data types in matched samples. Our strategy is to exploit the rich information derived from large scale population cohorts in the form of AF GWAS loci and the AF polygenic risk score in combination with our deep molecular phenotyping to identify putative AF core genes. We intentionally held out our cohort disease status to be able to use it as validation of our putative core genes in the end of the analysis.

The analysis of differentially expressed genes undisputedly is a very important tool, however, it is most suited to find the top strongest effects. Opposed to that, effect sizes for trans effects propagated through regulatory networks are often much smaller. Our enrichment based candidate selection approach largely reduces the search space to only few genes that are most promising to be trans-regulated.

Indeed, our case/control status data confirmed the disease link of our NKX2-5 targets as shown in **Figure 5e** in the last two columns, by the highly significant gene set enrichment analysis (GSEA) results for a collective downregulation of NKX2-5 targets for AF and we show the disease associations for all putative core genes in **Table 3** as well as in our new Figure 6a. The collective downregulation of NKX2-5 targets was also replicated in previously published transcriptome and proteome data sets.

In summary, our results go beyond differential gene expression analyses by providing data supporting a model that links a trans acting GWAS locus to several candidate core genes. Those candidate core genes are regulated through the transcription factor NKX2-5 by the rs9481842-NKX2-5 trans eQTL and show a highly significant collective downregulation in AF.

Figure 5e: NKX2-5 activity regulated by AF GWAS variant rs9481842.

Depicted are functional NKX2-5 targets with the number of TF binding sites (column 1), *trans* eQTL strength (columns 2-4), *trans* pQTL strength (columns 5-7) and protein level in AF (columns 8-9). The color scale represents median transcript or protein values per group (=columns). Residuals corrected for fibroblast-score and RIN-score / protein concentration with subsequent normal-quantile normalization per gene were used to calculate the medians per group. A quantitative description of the qualitative results presented in the heatmap can be found in Suppl Table S16 and Table 3.

Figure 6a: Replication of the core gene candidate AF association and NKX2-5 target coexpression in independent datasets.

Published proteomics data (PXD006675) as well as RNAseq data (GSE128188, GTEx) generated from human atrial tissue samples were used for replication.

a: Centered and scaled values of the mean mRNA or protein expression in AF ctrls and cases, with stronger effects on protein level. GSEA p-values quantify the negative association of NKX2-5 targets with respect to AF. Sample sizes per column: 69 controls, 14 prevalent AF cases, 69 controls, 14 prevalent AF cases (AFHRI, all right atrial appendage); five controls, five AF cases (GSE128188, both right atrial appendage); five controls, five AF cases (GSE128188, both left atrial appendage); 3 controls, 3 AF cases (PXD006675, both left atrium). A quantitative description of the qualitative results presented in the heatmap can be found in Suppl Table S13-S14 and Table 3.

AF, atrial fibrillation; Ctrl, control i.e. individuals in sinus rhythm; GSEA, gene set enrichment analysis; GTEx, Genotype-Tissue-Expression project; eQTL, expression quantitative trait loci; pQTL, protein quantitative trait loci. *Mutation known to affect cardiovascular phenotypes; **Mutation known to affect arrhythmias; *Differential expression functional impairment for cardiovascular phenotypes; **Differential expression or functional impairment for arrhythmias.

4. Could the small proportion of eQTL/pQTL overlap be due to differences in heritability between transcription and translation? A correlation of effect sizes or directions may better compare eQTLs and pQTLs given likely such systemic differences.

We agree with the reviewer that a more quantitative description of the differences between eQTL and pQTL is more telling than simple overlaps. That is why we had already included a supplementary figure comparing effect sizes between transcriptomics and proteomic QTLs in the first submission of the manuscript. We included further details in the manuscript and extended the corresponding figure and section in the supplement (Suppl Figure S5), to now also show all SNP-gene pairs and not just the top QTL per gene.

Compared to 8.2% of all significant SNP-gene pairs that are shared between mRNA and protein, much higher concordance can be observed when looking at correlation of effect sizes. We observe a Pearson correlation of 0.61 for effect sizes of all significant eQTLs and their corresponding pQTLs and a correlation of 0.79 for the effect sizes of all significant pQTLs when compared to the matching eQTLs. Storey's qvalue method estimates a replication rate of 20% for the top eQTLs in pQTLs and 32% for top pQTLs in eQTLs.

Figure S5: Between-OMIC comparison of *cis* quantitative trait loci results.

c: Comparison of effect sizes of eQTLs and pQTLs in the AFHRI cohort for all significant *cis* eQTL and *cis* pQTL SNP-gene pairs (FDR<0.05, Benjamini-Hochberg procedure) colored based on the functional QTL categories.

Still, the majority of loci is not shared in our data and also in other studies that compared eQTL and pQTL (for details refer to the Suppl Figure S5, the Suppl section “Correlation between mRNA and protein” where we compare correlations to the dataset of Battle et al. 2015, as well as the Suppl section “Overlap of *cis* eQTLs and pQTLs” with the **Suppl Table S2** with comparisons to Battle et al. 2015, Sun et al. 2018, Hause et al. 2014 and the GTEx data).

A number of mechanisms, such as post-transcriptional regulation or mRNA buffering could be responsible for these differences and thus variants affecting exclusively transcript or protein levels are of major interest for assessing the impact of genetics on these regulatory mechanisms. In particular, finding variants that exclusively affect protein levels demonstrates the added value of proteomic profiling to mRNA centric approaches. To do so, we have implemented two complementary approaches to identify truly independent QTL effects:

- 1) the regression based analysis that was already presented in the first version of the manuscript and
- 2) a formal colocalization analysis that we added in this revision.

To elaborate on this, we added **Suppl Figure S5c**, which illustrates the different clusters of QTLs: Shared eQTLs/pQTLs that are highly correlated and independent eQTLs that are most likely driven by genetic influences on transcription, and then the independent pQTLs that potentially arise by altering post-translational processes. So rather than differences in heritability of transcription and translation, we propose that genetic variants affect different stages of gene regulation based on the localization and consequence of a variant with respect to coding sequence and regulatory elements.

5. In Fig 3, I would add one or two non-cardiovascular traits as a negative control to verify tissue specificity and that general enrichment of all disease-associated variants isn't simply being observed.

Figure 3: Overlap of *cis* QTL associations with GWAS hits annotated in the GWAS catalogue.

a: Overview of significant *cis* eQTLs and pQTLs (FDR<0.05) overlapping with different disease traits.

b: Independent pQTL for GWAS hit creatine kinase levels. Shown are the significant *cis* eQTL, pQTL and ratio QTL for the SNP rs1801690 and the gene APOH (FDR<0.05). In the boxplots, the lower and upper hinges correspond to the first and third quartiles (the 25th and 75th percentiles). The median is denoted by the central line in the box. The upper/lower whisker extends from the hinge to the largest/smallest value no further than 1.5·IQR from the hinge.

c: For three different trait categories (cardiovascular traits, arrhythmias and atrial fibrillation) as well as rheumatoid arthritis as a negative control, the enrichment of GWAS hits at significant *cis* QTLs (FDR<0.05) was evaluated. Enrichments were calculated using Fisher's exact test (two-sided).

QTL, quantitative trait loci; GWAS, genome-wide association study; SNP, single-nucleotide polymorphism; eQTL, expression quantitative trait loci; pQTL, protein quantitative trait loci; ratio quantitative trait loci; CI, confidence interval; IQR, interquartile range.

We thank the reviewer for this suggestion. We have added rheumatoid arthritis to the figure, as there's no obvious shared disease mechanism with AF. As expected, we cannot find any rheumatoid arthritis loci that coincide with a significant eQTL or pQTL.

6. The authors assume that an AF PRS is a "proxy for an aggregation of AF-related trans effects across the whole genome." How are the authors excluding *cis*-acting variants associated with AF? With a full AF PRS, are they merely picking up strong *cis* effects?

We thank the reviewer for raising this very interesting point (also raised by reviewer 3, general comment 4). We have now adjusted our analysis strategy to explicitly rule out that *cis* effects on expression drive the association of genes with the PRS. For each transcript or protein, we included the strongest *cis* QTL SNPs as covariates when computing the correlation between a transcript or protein with the PRS.

The results of the new analysis confirm our previous findings. While there were small changes in the absolute number of enriched pathways and ranking of those, all GO terms mentioned in the manuscript before remain significant.

In summary, our results did not change qualitatively at all, indicating that our previously reported eQTLs/pQTLs associations were not driven by *cis* effects. Thanks to the reviewers question, we can now exclude this possible confounding.

7. Using summary-level data from sources such as GTEx, INTERVAL, SCALLOP, are the AF PRS effects on gene expression and protein concentrations described in the paper specific to atrial tissue or are such changes observed in other tissues?

To compute eQTS or pQTS rankings, the correlation between individual level transcript / protein abundance and the AF PRS is considered. It is not possible to derive this from summary-level data.

We can however assess general transcript and protein abundance differences of our putative core genes in different tissues using the GTEx RNAseq and proteomics data.

As expected, transcript expression patterns of the two heart tissues atrial appendage and left ventricle are very similar, followed by skeletal muscle (**Figure R3**). They are most distinct from whole blood and pancreas, but also don't share many similarities with the (coronary, aorta and tibial) artery tissues.

Similarly, protein abundance of the core genes is more common in heart and muscle compared to all other tissues.

For instance, we were able to identify similar expressions profiles between the heart and muscle for transcripts as well as proteins contributing to the contractile apparatus of both respective tissues (TNNC1, TCAP). The same holds true for the expression of the skeletal and cardiac muscle creatine kinase encoded by CKM, which was upregulated in both tissue on mRNA and protein level.

Figure R3: Tissue-specific transcript and protein expression of putative core genes in different GTEx tissues.

Minor:

1. I would suggest changing "multiOMICS" to "multiomics" since "OMICS" is not an acronym.

Thank you for this suggestion. The manuscript was amended accordingly.

2. cis and trans should be italicized.

Thank you for this suggestion. The manuscript was amended accordingly.

3. Gene names should be italicized.

Formatting guidelines suggest to have gene and protein names in normal font and only transcripts italicized.

4. "Coronary bypass" should be "coronary artery bypass." Also, it's noted that "patients with surgery other than coronary bypass, e.g. valve surgery, were excluded." Many patients undergo simultaneous coronary artery bypass and valve surgery; I assume these individuals are excluded? If so, please clarify in the text.

Yes, all individuals underwent coronary artery bypass surgery exclusively. Any patients with other or additional surgeries were excluded. We revised the manuscript accordingly. We now state in the methods: "Participants were older than 18 years of age and were scheduled to undergo open heart coronary artery bypass surgery. Any patients with other bypass surgeries or additional procedures, e.g. valve surgery, were excluded."

5. Methods indicate that direct genotypes were filtered for HWE $P > 1e-6$, and imputed genotypes for HWE $P > 1e-4$. Shouldn't HWE $P < 1e-6$ and $< 1e-4$ be removed?

Indeed, the wording might have been misleading. We have used the criterion for inclusion HWE $P > 1e-6$ (exclusion HWE $P = < 1e-6$) for directly measured genotypes and a more stringent cutoff after imputation: inclusion HWE $P > 1e-4$ (exclusion HWE $P = < 1e-4$). We now clearly state the inclusion thresholds in the manuscript.

6. What's the rationale for recoding SNPs with < 3 homozygous for the minor allele to heterozygous?

One or two data points (individuals) for the homozygous minor allele increase the risk that outliers in the expression data, which coincide with rare genotypes lead to false positive findings. To prevent spurious associations, they were recoded as described. We added the rationale in the methods to clarify the reason for this processing:

"For QTL analyses, outliers in the expression data which coincide with rare genotypes can lead to false positive findings. For SNPs with less than 3 individuals with the homozygous-minor-allele genotype, all samples with homozygous-minor-allele genotype were therefore recoded to heterozygous genotype."

Reviewer #2 (Remarks to the Author):

In this study, the authors propose an integrative analysis of genomics, transcriptomics and proteomics data of human atrial tissue to identify genome-wide genetic effects on intermediate molecular phenotypes in the context of AF.

Major comments

[1] While this is a fascinating study, the reviewer questions the exact novelty of the study in terms of mechanistic insight, and in particular, misses novel mechanistic data linking a specific gene / pathway to a specific mechanism contributing to AF. Such data would improve the manuscript considerably.

Thank you very much for this feedback. We would like to highlight that to the best of their knowledge, this is the first dataset analyzing the molecular consequences of common genetic variation on proteomics in human (atrial) tissue. By integrating genotype, transcript and protein data together with our functional QTL categories, we generate data driven hypothesis of molecular mechanisms (candidate genes) for GWAS loci for follow up. Our data thus strongly reduces the search space of potential causal mechanisms to evaluate. Of course, given the number of observations, experimental follow up on specific discoveries was not possible in the scope of this study.

More importantly, our trans QTL approach does establish novel links between AF GWAS loci with unknown functional mechanisms and the target transcript or protein.

Especially for the case of the rs9481842-NKX2-5 trans eQTL, we propose regulatory mechanisms how AF relevant target genes are regulated by the GWAS SNP through the transcription factor (TF) NKX2-5. We exploited heart specific prior data on NKX2-5 regulation and integrated this functional genomic data with our genome-wide transcriptomics data to derive TF activity of NKX2-5 in our patients. NKX2-5-specific binding sites were defined by integrating promoter annotations from Gencode, ChIPseq and promoter capture HiC data in human iPSC-derived cardiomyocytes as well as open chromatin regions based on chromatin states in heart atrial appendage tissue. This allowed us to globally quantify the impact of the trans acting AF SNP rs9481842 and to identify direct targets of NKX2-5 regulation. We further validated the rs9481842-NKX2-5 trans eQTL on proteomics level using Western blot analysis and observed a high correlation of the predicted TF activity with actual protein intensities (cor=0.42, P=0.026, two-sided test).

Included in this revision, we also replicated the novel

1. regulation of NKX2-5 targets by coexpression of NKX2-5 and the identified target transcripts in two independent RNAseq datasets and
2. disease association of the NKX2-5 targets with AF by the same collective downregulation of the 13 targets in patients with AF compared to ctrls using GSEA in an independent RNAseq and an independent proteomics dataset.

NKX2-5 itself has been known in the context of familial AF, where rare mutations contribute to disease risk. Our proposed mechanism however involves common genetic variation with much broader clinical implications. As opposed to affecting only a few cases with severe consequences, our thirteen identified targets can be further characterized for their potential as interventional targets. Our results provide the starting ground for further in depth studies evaluating the exact clinical implications for each of the identified genes.

Minor comments

[2] There is a lack of vital information in the Results section. This info can be found elsewhere in the manuscript, but its absence from the Results hinders reading. For example, the first paragraph of the Results misses important information concerning the tissue studied, it's origin, number of samples etc.

We would like to thank the reviewer for this suggestion and have amended the manuscript accordingly. We added as the first sentence for the result section: "Microarray transcriptomics and genome-wide proteomics were measured in human atrial tissue samples collected during coronary artery bypass surgery." As sample size differed for the different parts of analyses, those are now clearly stated in the corresponding tables.

[3] Similarly, for example, for Table 1. This needs to mention for example that this concerns atrial tissue and should mention the number of atrial samples studies.

Thank you for the feedback. We incorporated more information in the table captions (e.g. tissue types, cis or trans labels) or the table itself (e.g. sample sizes). **Table 1** now reads as follows:

Table 1: Summary of tested data and discovered *cis* quantitative trait loci.

Significant *cis* QTLs in human heart right atrial appendage tissue for mRNA and protein measurements for FDR <0.05 (according to Benjamini-Hochberg procedure) and P value <1×10⁻⁵. Loci denote the number of independent loci derived by LD-clumping.

QTL, quantitative trait loci; FDR, false discovery rate; LD, linkage disequilibrium; eQTL, expression quantitative trait loci; pQTL, protein quantitative trait loci; ratioQTL, ratio quantitative trait loci; N, sample size.

Results for all available transcriptomics and proteomics measurements:										
	Tested:			FDR<0.05:			P<1×10 ⁻⁵ :			
	SNPs	Pairs	Genes	Pairs	Genes	Loci	Pairs	Genes	Loci	N
eQTL	4 861 118	56 139 851	16 306	57 403	1 058	1 657	40 267	552	870	75
pQTL	2 323 504	4 508 654	1 337	4 081	91	139	2 543	45	71	75
Results only for genes with both transcriptomics and proteomics measurements:										
eQTL	2 249 758	4 198 168	1 243	4 603	124	201	3 218	64	109	75
pQTL	2 249 758	4 198 168	1 243	3 906	87	133	2 406	42	66	75
ratioQTL	2 249 758	4 198 168	1 243	563	16	23	575	18	27	66

[4] It is unclear whether SNPs of Table 2 correspond to SNPs from AF GWAS (or their proxies).

We generally used the original GWAS SNPs. However, some GWAS SNPs were not available in this cohort (rs9675122, rs34292822). In that case, we used one best proxy (rs11658168, rs11588763) instead. This is now clearly stated in the table:

Table 2: *Trans* QTL results.

Significant *trans* eQTLs and pQTLs for a FDR<0.2 (Benjamini-Hochberg procedure). *Trans* analyses were performed on 23 transcripts with 74 samples and 152 proteins with 73 samples of human heart right atrial appendage tissue for 108 variants associated with atrial fibrillation from the GWAS catalog (or their proxy, if the GWAS SNP was not measured). Calculations were carried out using the SNP rs11658168 as a proxy for the GWAS SNP rs9675122 as well as rs11588763 instead of the GWAS SNP rs34292822.

eQTL, expression quantitative trait loci; pQTL, expression quantitative trait loci; FDR, false discovery rate; GWAS, genome-wide association study; SNP, single- nucleotide polymorphism.

*Mutation known to affect cardiovascular phenotypes; **Mutation known to affect arrhythmias; †Differential expression functional impairment for cardiovascular phenotypes; ††Differential expression or functional impairment for arrhythmias; For details to disease links in literature see Suppl Table S12.

GWAS SNP	Variant		Position	Gene		QTL	β	trans QTL		
	QTL SNP	Chr		Symbol	Chr			T value	P value	FDR
rs9675122	rs11658168	chr17	7 406 134	TNNT2 * ^{††}	chr1	transcript	-0.517	-4.27	6.43×10^{-5}	0.0812
rs9481842	rs9481842	chr6	118 974 798	NKX2-5 **	chr5	transcript	-0.593	-4.27	6.54×10^{-5}	0.0812
rs34292822	rs11588763	chr1	154 813 584	CYB5R3	chr22	protein	-0.786	-4.89	6.86×10^{-6}	0.113
rs34292822	rs11588763	chr1	154 813 584	NDUFB3 [†]	chr2	protein	-0.916	-4.44	3.56×10^{-5}	0.133
rs9675122	rs11658168	chr17	7 406 134	HIBADH	chr7	protein	-0.512	-4.43	3.66×10^{-5}	0.133
rs34292822	rs11588763	chr1	154 813 584	NDUFA9 ^{††}	chr12	protein	-0.752	-4.42	3.85×10^{-5}	0.133
rs34292822	rs11588763	chr1	154 813 584	DLAT	chr11	protein	-0.716	-4.40	4.05×10^{-5}	0.133

[5] Data, Figure 2b and 2c: can you please clarify that lack of detection of both eQTL and pQTL was not due to a statistical power issue in the corresponding dataset?

There are three very important reasons for a lack of detection of both eQTL and pQTL:

Different coverage of genes and proteins measured:

The number of measured transcripts and proteins (transcripts: 16 306, proteins: 1 337, overlap: 1 243) differed significantly. Therefore, when looking at the overlap of eQTLs and pQTLs, we always restricted our analysis to those genes that were quantified in both omics.

QTLs with small effect sizes can be missed in a low power setting:

The power to detect QTLs mainly depends on effect size and sample size. For the *cis* eQTL and pQTL analysis, sample size was exactly the same. Battle et al. (2015) examined more deeply, how effect sizes of *cis* QTLs differed for transcriptomics, ribosomal profiling and proteomics data and came to the conclusion, that measurement noise played a rather neglectable role compared to the actual regulatory mechanisms. However, they observed on average reduced effect sizes of protein QTL compared to eQTL or ribosome occupancy QTL. So these smaller effect sizes could lead to lower power and explain a part of the observations.

QTLs are actually independent:

The reduced average effect sizes on protein level observed by Battle et al. (2015) were also driven by cases where QTL effects were completely absent on protein levels, suggesting truly independent regulatory mechanisms. Post-transcriptional regulation or mRNA buffering can be responsible for genetic variation that affects transcript or protein levels exclusively. As a result, next to shared eQTLs/pQTLs that are highly correlated we can observe independent eQTLs that are most likely driven by genetic influences on transcription and independent pQTLs that potentially arise by altering post-translational processes.

We have implemented two complementary approaches to distinguish truly independent genetic effects from cases where statistical power was not high enough to discover shared effects:

- 1) the regression based analysis that was already presented in the first version of the manuscript and
- 2) a formal colocalization analysis that we added in this revision.

The residual regression approach and the formal colocalization analyses agreed on the functional category in around two thirds of cases, with the colocalization showing a tendency to be less strict as opposed to the residual regression approach. While both methods detected functional QTL the other did not, both consistently confirmed the existence of those three distinct categories.

Taken together, we propose that genetic variants affect different stages of gene regulation based on the localization and consequence of a variant with respect to coding sequence and regulatory elements.

Reviewer #3 (Remarks to the Author):

The identification of over 100 loci associated with atrial fibrillation (AF) has enabled the generation of genome-wide polygenic risk scores (PRS) for AF risk. While rare monogenic forms of familial AF carry a high risk, the additive effects of many common genetic variants explain a much greater proportion of AF risk in the population. However, the precise mechanisms by which AF-associated single nucleotide polymorphisms (SNP) modulate the expression of multiple genes and pathways remains unclear especially as >95% of the polymorphisms occur in non-coding regions. Assessing tissue-specific cis-acting expression quantitative trait loci (eQTL) is a commonly used approach to decipher the SNP-gene relationship but this strategy explains only a small fraction of the identified AF risk loci. However, the contributions of trans-genetic variants, distant to the target gene, to the genetic architecture of complex polygenic traits are unknown. According to the theoretical omnigenic model, trans-effects may explain up to 70% of the heritability of a complex disease, like AF, in part through gene regulatory networks. The goals of this study were to establish methods to identify core AF genes targeted by gene regulatory networks, and to integrate genomic, transcriptomic, and proteomic data from human atrial tissue to better understand how the relationship between common genetic variants and molecular changes in AF and uncover genotype-phenotype relations. Assum and colleagues identified two trans eQTLs and 4 trans protein (p) QTLs for the GWAS SNP rs9481842 and AF. Collectively, this study provides a comprehensive multi-omics approach to identifying trans-acting networks in small datasets and provides a rich resource of atrial tissue-specific regulatory variants for transcript and protein levels for gene prioritization.

General Comments

Overall, Assum and colleagues propose a novel multi-omics approach that integrated genomics, transcriptomics, and proteomics data from human atrial tissue to identify the underlying molecular networks for AF and uncover genotype-phenotype relationships. The manuscript is well-written and the conclusions on the whole are supported by the data; the methods are highly innovative especially a novel targeted trans QTL approach based on PRS; the findings are intriguing and may potentially provide important insights into molecular mechanisms by which AF SNPs modulate genes and signaling pathways using both cis- and trans-genetic effects. The statistical analysis appears to be appropriate and sufficient detail is provided to reproduce the work. Addressing the following specific comments will greatly aid the reader in the interpretation of the findings and potential impact of the study.

We would like to thank the reviewer for highlighting that our study is highly innovative. We agree with the reviewer that the suggested additional analysis greatly aid in the interpretation of the findings, in particular we provide more details on the comparison of eQTL and pQTL and we rule out confounding of the association of expression and the PRS by cis acting genetic factors.

Specific Comments

1. The additional 'PEER factors to adjust for known as well as unknown confounders' should be described in the main manuscript.

We appreciate the reviewers interest in the inferred PEER factors. PEER is a method commonly used to adjust high dimensional data for potential confounding (unobserved) latent factors. It is related to factor analysis and works in an unsupervised way. The resulting factors capture variability in many expression levels and can be interpreted similarly to the results of a PCA. To gain insights into what variation is actually captured by the unsupervised analysis, we visualize the relation of the PEER factors with measured covariates in Suppl Figure S2. This comparison shows that PEER factors are correlated with measured covariates, indicating that they captured variability caused by known factors (such as “disease status, cardiovascular risk factors and technical covariates” as we write in the manuscript). In addition some PEER factors show lower correlation with measured covariates and therefore capture unknown confounding factors. This kind of analysis is common practice in large-scale QTL studies and in our opinion does not provide additional biological insights. Therefore we would refrain from extending the description in the main text, unless the reviewer thinks more details on which covariates are actually correlated with PEER factors would enhance the results presented here.

Figure S2: Correlation of PEER factors with common risk factors of AF and technical covariates.

Pearson correlation of different PEER factors and known risk factors or technical covariates.

a: Transcriptomics analysis: Fibroblast-score and RIN-score highly correlate with PEER factors used in the final *cis* eQTL analysis.

b: Proteomics analysis: fibroblast-score and original sample protein concentration highly correlate with PEER factors used in the final *cis* pQTL analysis.

PEER, probabilistic estimation of expression residuals; QTL, quantitative trait loci; AF, (prevalent) atrial fibrillation; RIN, RNA integrity number; BMI, body mass index; diasBP, diastolic blood pressure; sysBP, systolic blood pressure; HF, heart failure; MI, myocardial infarction; CRP, C-reactive protein; NT-proBNP, N-terminal prohormone of brain natriuretic peptide.

2. The replication rate of eQTLs in GTEx atrial appendage tissue was assessed. While the 'effect sizes for the best eQTL ($P < 1 \times 10^{-5}$) per gene showed a correlation of 0.83 ($P = 3.6 \times 10^{-67}$) in GTEx, 66% replicated (GTEx $P < 1 \times 10^{-5}$) and 88% showed concordant allelic effects', what was the correlation for the least concordant eQTLs?

To show a more complete picture of the overlap of our eQTLs with the data provided by the GTEx consortium, we extended the corresponding Suppl Figure S3b to show all SNP-gene pairs that were significant in one of the datasets. We improved our mapping of genes between the cohorts, which led to small differences in numbers. When considering all SNP-gene pairs and not just the top eQTL per gene, correlation of effect sizes was slightly higher with 0.83 (compared to updated 0.81, $P = 2.3 \times 10^{-71}$) and only 14% of QTLs showed discordant allelic effects. Again, none of those 14% of SNP-gene pairs with differing signs of effect sizes were significant in the GTEx data ($P < 10^{-5}$, smallest $P = 0.0039$). Both analyses show that overall the cis eQTL are highly reproducible between the two data sets.

Figure S3: Comparison of *cis* eQTL and pQTL results to GTEx *cis* eQTLs in atrial appendage tissue.

a: Comparison of effect sizes of eQTLs in the GTEx and AFHRI cohort for i) the top significant *cis* eQTL per gene in GTEx ($P < 1 \times 10^{-5}$) and ii) the top significant *cis* eQTL per gene in AFHRI ($P < 1 \times 10^{-5}$).

b: Comparison of effect sizes of eQTLs in the GTEx and AFHRI cohort for i) all significant *cis* eQTL SNP-gene pairs in GTEx ($P < 1 \times 10^{-5}$) and ii) all significant *cis* eQTL SNP-gene pairs in AFHRI ($P < 1 \times 10^{-5}$).

c: Comparison of effect sizes of eQTLs in GTEx and pQTLs in the AFHRI cohort i) the top significant *cis* eQTL per gene in GTEx ($P < 1 \times 10^{-5}$) and ii) the top significant *cis* pQTL per gene in AFHRI ($P < 1 \times 10^{-5}$).

eQTL, expression quantitative trait loci; pQTL, protein quantitative trait loci; GTEx, Genotype-Tissue Expression project.

3. It is a little surprising that 'only 8.2% of significant SNP-gene associations are shared between mRNA and proteins (Figure 1, Suppl Figure S5, Suppl Table S2)'. The potential explanations for such a low shared rate between mRNA and proteins should be discussed especially with many studies using transcriptomic analyses to infer SNP-gene associations.

We fully agree with the reviewer that the low overlap of eQTLs and pQTLs has important implications. As specific questions regarding the eQTL/pQTL overlap were also raised by other reviewers, we extended the corresponding sections in the manuscript and supplement to add more quantitative descriptions of the overlap.

When looking at correlation of effect sizes, they show a much higher concordance (see also response to reviewer 1 comment 4). We observe a Pearson correlation of 0.61 for effect sizes of all significant eQTLs and their corresponding pQTLs and a correlation of 0.79 for the effect sizes of all significant pQTLs when compared to the matching eQTLs. Storey's qvalue method estimates a replication rate of 20% for the top eQTLs in pQTLs and 32% for top pQTLs in eQTLs.

Figure 2: Different genetic regulatory patterns derived by multiomics *cis* QTL integration.

a: Shared eQTLs / pQTLs represent QTLs, where the effect of transcriptional regulation translates into mRNA and protein abundance exemplified by the significant SNP - gene pair rs9664184 - MYOZ1. No corresponding ratio QTL can be observed as the genetic variation is shared across both omics levels.

b: Independent eQTLs depict variants with regulation on mRNA but not on protein level displayed by the significant SNP - transcript pair rs2070594 - ATP5C1.

c: Independent pQTLs represent variants that show regulation only on protein level as shown for the SNP-protein pair rs3916 - ACADS. Genetic influence is not observable on transcript level.

In the boxplots, the lower and upper hinges correspond to the first and third quartiles (the 25th and 75th percentiles). The median is denoted by the central line in the box. The upper/lower whisker extends from the hinge to the largest/smallest value no further than 1.5 · IQR from the hinge.

eQTL, expression quantitative trait loci; pQTL, protein quantitative trait loci; ratioQTL, ratio quantitative trait loci; TssA, active transcription start site; UTR, untranslated region; TF BS, transcription factor binding site; RBP, RNA binding protein; SNP, single-nucleotide polymorphism; IQR, interquartile range.

Still the majority of loci remains not shared in our data. That is why we confirmed the low rate of concordance in other datasets (see Supplement “Overlap of *cis* eQTLs and pQTLs”) as well. More importantly, we had introduced our functional QTL categories (shared / independent, see also Figure 2, where we added the corresponding p values) to also distinguish truly independent genetic effects from cases where statistical power was not high enough to discover shared effects in both mRNA and protein levels:

A number of mechanisms, such as post-transcriptional regulation or mRNA buffering could be responsible for these differences and thus variants affecting exclusively transcript or protein levels are of major interest for assessing the impact of genetics on these regulatory mechanisms. In particular, finding variants that exclusively affect protein levels demonstrates the added value of proteomic profiling to mRNA centric approaches. To do so, we have implemented two complementary approaches to identify truly independent QTL effects:

- 1) the regression based analysis that was already presented in the first version of the manuscript and
- 2) a formal colocalization analysis that we added in this revision.

To elaborate on this, we added **Suppl Figure S5c**, where can clearly observe different cluster of QTLs: Shared eQTLs/pQTLs that are highly correlated and independent eQTLs that are most likely driven by genetic influences on transcription, and then the independent pQTLs that potentially arise by altering post-translational processes. To shed light on the potential mechanisms we had performed an enrichment analysis for the top SNPs in the different QTL categories (**Suppl Figure S7a**), showing different sequence elements enriched in independent eQTL (TFBS) or pQTL (exons). However due to the small number of QTLs falling into each category this analysis was not highly powered and we cannot currently pinpoint the exact regulatory mechanisms affected by sequence variants.

Taken together, we propose that genetic variants affect different stages of gene regulation based on the localization and consequence of a variant with respect to coding sequence and regulatory elements.

Figure S5: Between-OMIC comparison of *cis* quantitative trait loci results.

c: Comparison of effect sizes of eQTLs and pQTLs in the AFHRI cohort for all significant *cis* eQTL and *cis* pQTL SNP-gene pairs (FDR<0.05, Benjamini-Hochberg procedure) colored based on the functional QTL categories.

With respect to the implications we conclude in the discussion: “In line with prior studies, we observed large differences in transcript and protein expression as well as their regulation,^{14,15,20} emphasizing the necessity and benefit of taking multiple molecular entities into account to investigate genotype-phenotype relationships.”

4. The investigators used the PRS as a proxy for an aggregation of AF-related trans-effects across the whole genome. However, many of the AF-associated SNPs incorporated into PRS regulate nearby genes, i.e., display cis-genetic effects. Thus, it is unclear why both cis- and trans-effects are included in the model.

We thank the reviewer for raising this very interesting point (also raised by reviewer 1). We have now adjusted our analysis strategy to explicitly rule out that cis effects on expression drive the association of genes with the PRS. For each transcript or protein, we included the strongest cis QTL SNPs as covariates when computing the correlation between a transcript or protein with the PRS.

The results of the new analysis confirm our previous findings. While there were small changes in the absolute number of enriched pathways and ranking of those, all GO terms mentioned in the manuscript before remain significant.

In summary, our previous results were including both cis and trans effects. The new version is now based on trans effects exclusively. This did not qualitatively change our results at all, indicating that our previously reported eQTS/pQTS associations were not driven by cis effects. Thanks to the reviewers question, we can now exclude this possible confounding.

5. The link between the core candidates genes and AF was established using either trans eQTLs or pQTLs for the AF GWAS hits. Surely it would be better to use the 21 genes with mRNA and protein concordance?

We absolutely agree with the reviewer that this kind of action would have been the most obvious choice. However, if we would like to express one key message by the first part of the manuscript, this would be that there are vast differences in regulation for the different kinds of omics. The choice to consider different candidates for trans eQTL and trans pQTL testing was further reinforced by the differing coverage of genes measured. Out of 23 trans eQTL candidates that have been identified using the updated analyses correcting for cis effects, only 14 were measured on protein level.

Nevertheless, especially for a transcription factor like NKX2-5, genetic regulation is more likely to be functional in case of a significant effect on protein level. Nevertheless, this protein was not captured by the proteome wide survey. So we resorted to a targeted Western blot analysis, which successfully replicated the NKX2-5 eQTL. We indeed confirmed a significant NKX2-5 pQTL for 29 samples of the original cohort where remaining tissue was available. We would like to stress again, that we could not have found and replicated this trans association, if we had restricted ourselves to overlapping eQTL/pQTL from the beginning.

6. The number of right atrial tissue samples used for trans-QTL analyses is unclear and should be clearly stated in the manuscript. It is also important to discuss the limitations of using right atrial appendage tissue samples to investigate the association between common genetic variants and regulatory genes for AF.

We apologize for the insufficient information given in the manuscript and have amended the table description accordingly:

Table 2: *Trans* QTL results.

Significant *trans* eQTLs and pQTLs for a FDR<0.2 (Benjamini-Hochberg procedure). *Trans* analyses were performed on 23 transcripts with 74 samples and 152 proteins with 73 samples of human heart right atrial appendage tissue for 108 variants associated with atrial fibrillation from the GWAS catalog (or their proxy, if the GWAS SNP was not measured). Calculations were carried out using the SNP rs11658168 as a proxy for the GWAS SNP rs9675122 as well as rs11588763 instead of the GWAS SNP rs34292822.

eQTL, expression quantitative trait loci; pQTL, expression quantitative trait loci; FDR, false discovery rate; GWAS, genome-wide association study; SNP, single- nucleotide polymorphism.

*Mutation known to affect cardiovascular phenotypes; **Mutation known to affect arrhythmias; *Differential expression functional impairment for cardiovascular phenotypes; **Differential expression or functional impairment for arrhythmias; For details to disease links in literature see Suppl Table S12.

GWAS SNP	Variant		Position	Gene		QTL	β	trans QTL		
	QTL SNP	Chr		Symbol	Chr			T value	P value	FDR
rs9675122	rs11658168	chr17	7 406 134	TNNT2 * ⁺⁺	chr1	transcript	-0.517	-4.27	6.43×10^{-5}	0.0812
rs9481842	rs9481842	chr6	118 974 798	NKX2-5 **	chr5	transcript	-0.593	-4.27	6.54×10^{-5}	0.0812
rs34292822	rs11588763	chr1	154 813 584	CYB5R3	chr22	protein	-0.786	-4.89	6.86×10^{-6}	0.113
rs34292822	rs11588763	chr1	154 813 584	NDUFB3 ⁺	chr2	protein	-0.916	-4.44	3.56×10^{-5}	0.133
rs9675122	rs11658168	chr17	7 406 134	HIBADH	chr7	protein	-0.512	-4.43	3.66×10^{-5}	0.133
rs34292822	rs11588763	chr1	154 813 584	NDUFA9 ⁺⁺	chr12	protein	-0.752	-4.42	3.85×10^{-5}	0.133
rs34292822	rs11588763	chr1	154 813 584	DLAT	chr11	protein	-0.716	-4.40	4.05×10^{-5}	0.133

To date, multiple studies have used atrial appendages to analyze and identify relevant disease-mechanisms or candidate genes on multiple molecular levels (Mayr et al., 2008; Roselli et al., 2018, Censi et al., 2010; Martin et al., 2015; Brundel et al., 2001). It is the tissue available for research since the pulmonary vein ostia cannot be used for safe biopsies. Furthermore, although AF in many cases originates from the pulmonary vein ostia, other mechanisms like electrical and structural remodeling of the atrial myocardium can represent substrates for AF initiation. Therefore, we believe that the extracted tissue samples are well suited and the best proxy for atrial impairment for our analysis. The use of this type of tissue renders our results comparable to prior research and will permit comparable research for validation.

We address this now in the discussion: “Furthermore, although AF prevalently originates from pulmonary vein ostia, the relevance and usefulness of right atrial appendage tissue was demonstrated by prior studies which identified various AF disease-mechanisms and candidate genes.^{3,46,47} Therefore, we believe that the extracted tissue samples are well suited and the best proxy for atrial impairment for our analysis. Another strength worth mentioning is that tissue samples were explanted during cardiac surgery and not post-mortem as in comparable datasets, which can affect various pathways e.g. metabolism.”

The relevance of right atrial appendage tissue for AF is further demonstrated by the clear enrichment of GWAS hits at significant eQTL and pQTL loci in our study. This enrichment was the strongest for AF GWAS hits compared to other GWAS studies for arrhythmias in general or other cardiovascular diseases (e.g. coronary artery disease).

7. The weak association between SNP rs8481842 and transcription factor (TF) is troubling as is the failure to detect NKX2-5 at the protein level in atrial tissue.

The association of the AF GWAS SNP rs9481842 with mRNA expression of the TF NKX2-5 was highly significant with a nominal P value of 6.54×10^{-5} and we now show new data that confirm the trans eQTL on protein level.

Due to its low abundance which is generally in line with other transcription factors, NKX2-5 is difficult to detect using mass spectrometry experiments. Comparable datasets, e.g. PXD006675 for the left atrium and proteomics of the GTEx consortium in the right atrial appendage, also failed to detect NKX2-5. Nevertheless, we validated our findings experimentally on protein level using Western blot analysis and succeeded in quantifying NKX2-5 in a limited number of samples, where remaining tissue was available.

Following up on this, we also replicated the regulation of our 13 identified NKX2-5 targets by coexpression as well as the AF disease link of these targets by GSEA in two independent datasets each, which we would like to address in more detail for your following comment on replication in independent datasets.

1. rs9481842 - NKX2-5 trans pQTL:

Using Western blot analysis, we were able to quantify NKX2-5 abundance in 29 remaining tissue samples. Evaluating the rs9481842 - NKX2-5 trans pQTL similar to the trans eQTL showed a significant association of protein intensities with the SNP rs9481842 ($\beta = -0.45$, $t = -2.9$, $P = 0.049$). Sex was dropped as a covariate because of only one female sample being available for this analysis.

Figure 5b-5d: NKX2-5 activity controlled by AF GWAS variant rs9481842.

b: Strong *trans* eQTL association between the SNP rs9481842 and the NKX2-5 transcript.

c: Validation of the NKX2-5 trans eQTL on protein level (trans pQTL) using Western blot analysis in remaining tissue samples.

d: NKX2-5 activity estimation based on target mRNA expression stratified by the rs9481842 genotype.

2. High correlation of NKX2-5 protein intensity and estimated NKX2-5 TF activity:

We used genome-wide transcriptomics data as well as additional independent tissue and cell type specific annotations to estimate NKX2-5 TF activity as a proxy for protein measurements. Having now access to the Western blot analyses, we compared the actual protein abundance to the predicted TF activity and observed a strong Spearman rank correlation of 0.42 ($P = 0.026$, two-sided test, **Suppl Figure S12**).

8. There are a number of limitations of the study that should be discussed fully:
- i) The small clinical dataset, and sample size and the heterogeneity of the underlying substrate for AF.

We agree with the reviewer, that these are important points to discuss. We have added content on the relevance of atrial appendage for AF and now state in the corresponding section of the discussion:

"We acknowledge some limitations that are attributed to common biological and technical factors. First, the use of human heart tissue came with several challenges including restricted sample sizes and heterogeneity of cellular composition. The small sample size affects the statistical power of QTL analyses and does not allow for assessing causality of molecular and physiological changes, for example by Mendelian randomization.^{41,42} Changes in cell-type composition and structural remodeling have been described for the pathology of AF.⁴³ To take differences in the cellular composition into account, we used a fibroblast-score based on a fibroblast-specific gene signature to adjust expression levels in eQTL/pQTL analyses.⁴⁴ Furthermore, although AF prevalently originates from pulmonary vein ostia, the relevance and usefulness of right atrial appendage tissue was demonstrated by prior studies which identified various AF disease-mechanisms and candidate genes.^{3,45,46} Therefore, we believe that the extracted tissue samples are well suited and the best proxy for atrial impairment for our analysis. Another strength is that tissue samples were explanted during cardiac surgery and not post-mortem as in other datasets like GTEx¹⁵, which can affect various pathways e.g. metabolism."

The main goal of our new PRS based pathway enrichment approach for candidate gene selection is to make trans QTL analyses feasible given the aforementioned limitations.

For the cis QTL analyses, the use of PEER factors instead of known covariates further enables us to correct for unknown confounders.

Finally, replication of our results wherever it was possible demonstrates that a certain reproducibility is given.

- ii) The GO gene sets were selected and failed to include calcium homeostasis, a key signaling pathway in the pathogenesis of AF.

Thank you for highlighting this. Our original analysis indeed had identified GO terms related to calcium homeostasis, which we had previously reported in the **Suppl Table S7**. We have now also mentioned this result in the main text:

"Furthermore, we identified three processes that implicate calcium homeostasis ("GO:0010880 Regulation of release of sequestered calcium ion into cytosol by sarcoplasmic reticulum", "GO:0010881 Regulation of cardiac muscle contraction by regulation of the release of sequestered calcium ion" and "GO:0010882 Regulation of cardiac muscle contraction by calcium ion signaling")."

We intentionally used the GO biological processes for this analysis, which are a priori not related to specific diseases (as opposed to KEGG disease pathways), to avoid introducing bias which could lead to circular reasoning.

- iii) Replication in an independent dataset was not performed.

Despite our best efforts, replication of the cis pQTLs, trans eQTLs and trans pQTLs results in an independent dataset was not possible, as such datasets were not accessible to the authors. Instead, we replicated the NKX2-5 trans eQTL on protein level using new independent measurements in our cohort (response to your comment 7) and focused on the

downstream effects of NKX2-5 and the 13 identified targets. More than half of our putative core genes were already mentioned by other studies in the context of AF, arrhythmias or other cardiovascular disease. We present those finding in **Suppl Table S12**. to We further successfully replicated the coordinated downregulation of the targets in AF and the coexpression of NKX2-5 and its targets in two independent datasets each:

AF association of 13 identified NKX2-5 targets (two independent datasets)

We showed a consistent collective downregulation of the 13 NKX2-5 targets in patients with prevalent AF compared to controls using GSEA analysis ($P=7.17 \times 10^{-5}$). Repeating the analysis for the proteomics data set PXD006675 showed a comparable negative enrichment of our NKX2-5 target gene set of $P=1.24 \times 10^{-4}$, and we further observed a negative enrichment of $P=0.0189/P=0.00141$ in right/left atrial appendage samples of the RNAseq dataset GSE128188.

Coexpression of NKX2-5 with the 13 identified targets (two independent datasets)

Another strong validation of the regulatory link between NKX2-5 and the identified targets was a high coexpression of NKX2-5 and its target transcripts.

Pearson's correlation of the NKX2-5 transcript with the 13 target transcripts ranged between 0.43 and 0.70 (median: 0.54) in the GTEx (right) atrial appendage tissue.

Similarly high correlations were observed in right (0.16-0.81, median: 0.58) and left (0.54-0.87, median: 0.72) atrial appendage samples of the RNAseq dataset GSE128188.

The new results are presented in the added **Figure 6**.

Figure 6: Replication of the core gene candidate AF association and NKX2-5 target coexpression in independent datasets.

Published proteomics data (PXD006675) as well as RNAseq data (GSE128188, GTEx) generated from human atrial tissue samples were used for replication.

a: Centered and scaled values of the mean mRNA or protein expression in AF ctrls and cases, with stronger effects on protein level. GSEA p-values quantify the negative association of NKX2-5 targets with respect to AF. Sample sizes per column: 69 controls, 14 prevalent AF cases, 69 controls, 14 prevalent AF cases (AFHRI, all right atrial appendage); five controls, five AF cases (GSE128188, both right atrial appendage); five controls, five AF cases (GSE128188, both left atrial appendage); 3 controls, 3 AF cases (PXD006675, both left atrium). A quantitative description of the qualitative results presented in the heatmap can be found in Suppl Table S13-S14 and Table 3.

b: Coexpression of NKX2-5 with the 13 identified NKX2-5 transcription factor targets (Pearson's correlation). Quantified is the correlation between NKX2-5 and its targets on mRNA level for mRNA datasets and the correlation between the NKX2-5 transcript expression with the target protein concentrations for the AFHRI proteomics (NKX2-5 not quantified in proteomics). Sample sizes used for

the computation of correlations: 102 AFHRI mRNA , 96 AFHRI protein, 372 GTEx, ten GSE128188 right, and ten left atrial appendage samples.

AF, atrial fibrillation; Ctrl, control i.e. individuals in sinus rhythm; GSEA, gene set enrichment analysis; GTEx, Genotype-Tissue-Expression project; eQTL, expression quantitative trait loci; pQTL, protein quantitative trait loci. *Mutation known to affect cardiovascular phenotypes; **Mutation known to affect arrhythmias; †Differential expression functional impairment for cardiovascular phenotypes; ††Differential expression or functional impairment for arrhythmias.

9. Minor:

i) Inconsistent use of abbreviations – too many to list.

We appreciate the feedback and have amended the manuscript to the best of our knowledge.

ii) The concept of different genetic regulatory patterns derived by multi-omics QTL integration is unclear and requires clarification.

Our study provided the unique possibility of directly integrating individual level genotypes, transcript and protein measurements, which enabled us to use a regression based analysis to distinguish truly independent genetic effects from cases where statistical power was not high enough to discover shared effects in both mRNA and protein levels.

For every SNP, a residual eQTL considers the association between a variant and the mRNA measurements after removing variation shared with protein measurements for the same gene. Vice versa, a residual pQTL describes the association between the variant and protein levels after removing variation shared with mRNA. Having removed variation that was shared across omics levels for each gene, we can now assess the omic-specific effect of a variant.

We therefore defined the following three functional categories:

1. Shared eQTL / pQTL:
Both the residual eQTL and residual pQTL should disappear when looking at a shared QTL.
2. Independent eQTL:
Conversely, independent eQTLs show a significant eQTL and residual eQTL without pQTL or residual pQTL.
3. Independent pQTL:
We observe a significant pQTL and residual pQTL but neither an eQTL nor residual eQTL.

We now state in the manuscript:

“Lack of overlap can either indicate independent effects or lack of power in one of the analyses. To distinguish these scenarios, we quantified for every variant if its effect was shared between the omic levels using a linear regression based residual QTL analysis. A residual eQTL considers the association between a variant and the mRNA measurements after removing variation shared with protein measurements for the same gene. Vice versa, a residual pQTL describes the association between the variant and protein levels after removing variation shared with mRNA. Having removed variation that was shared across omics levels for each gene, we can now assess the omic-specific effect of a variant.”

As mentioned before, this regression based approach is now complemented with a formal colocalization analysis.

iii) Figure 2 should include P-values.

We thank you for your suggestion and have amended **Figure 2** (shown above) and **Suppl Figure S6** accordingly.

Reviewer #4 (Remarks to the Author):

I read with interest the article by Assum et al. on multi-omic analyses for atrial fibrillation. In this paper, the authors collect phenotypic, genotyping, transcriptomic and proteomic data from human heart atrial tissue from ~100 individuals undergoing coronary bypass surgery and jointly analyze the data to make inference regarding genes that are related to atrial fibrillation by acting on human atrial tissue. In doing so, they also create a resource of atrial eQTLs and pQTLs which they make publicly available. The study is generally well designed and executed although some methodological aspects need to be improved/clarified. In addition, power/sample size limitations dampen enthusiasm in the absence of independent validation.

Major concerns:

1. The authors undertake an interesting approach to identify “core genes” for atrial fibrillation by leveraging an eQTS approach followed by a targeted trans-QTL analysis. The design of the approach is reasonable but the trans-QTL associations remain underpowered (no association has an FDR<0.05 even within this limited multiple testing burden). Consequently, it is necessary to provide evidence of in independent samples.

Thank you for this important remark. In this study we have generated a distinct dataset with deep molecular phenotyping, which integrates genotypes, transcript and protein measurements in human atrial tissue in a case control cohort of atrial fibrillation. Unfortunately, this study design is unique as is the generated data. Therefore, the analysis cannot be replicated in an independent cohort. We describe in the discussion further details on the uniqueness of the AFHRI dataset. However, we believe that our results are backed both with additional results we obtained in our cohort, as well as additional external replications of the disease associations of the identified genes.

Figure S12a-S12b: Inferred transcription factory activity strongly correlates with protein intensity.

a: *Trans* eQTL of the SNP rs9481842 and the transcription factor *NKX2-5* discovered by our polygenic risk score based enrichment approach for *trans* QTL gene candidate selection using microarray mRNA quantifications.

b: *Trans* pQTL validation of the rs9481842-*NKX2-5* *trans* eQTL in remaining tissue samples using Western blot analysis.

As additional strong support for the rs9481842-NKX2-5 trans eQTL, we successfully replicated this finding on protein level. Using Western blot analysis, we were able to quantify NKX2-5 abundance in 29 remaining tissue samples (**Suppl Figure S9, Suppl Figure S12**).

rs9481842 - NKX2-5 trans pQTL

Evaluating the rs9481842 - NKX2-5 trans pQTL similar to the trans eQTL showed a significant association of protein intensities with the SNP rs9481842 ($r=-0.45$, $t=-2.9$, $P=0.049$). Protein expression levels were also highly correlated with inferred NKX2-5 activity (Spearman rank correlation of 0.42, $P=0.026$, two-sided test). These results based on independent measurements are strong support for a true trans association in our cohort.

Internal and external replication of disease relevance of AF core genes

One of the key properties of AF core genes is their direct association with disease. More than half of our putative core genes have been reported in independent studies in the context of arrhythmias and other cardiovascular diseases as we report in detail in **Suppl Table S12**. NKX2-5 mutations have been described to cause AF and other arrhythmias and changes in NKX2-5 binding can contribute to electrocardiographic phenotypes. The TNNT2 protein was also found to be differentially expressed in the left atrium in AF and TNNT2 mutations are known to cause dilated cardiomyopathy. Also NDUFA9 and NDUFB3 have been reported in the context of AF and cardiomyopathy respectively.

We had previously presented additional internal validation of the association of 13 identified NKX2-5 targets with AF: we showed a consistent collective downregulation of the 13 NKX2-5 targets in patients with prevalent AF compared to controls using GSEA analysis ($P=7.17\times 10^{-5}$).

Now we replicate these findings in two additional independent datasets on protein and transcript expression level. Repeating the analysis for the proteomics data set PXD006675 (differential abundance results added in **Suppl Table S14**) showed a comparable negative enrichment for our NKX2-5 target gene set of $P=1.24\times 10^{-4}$ in left atrial tissue samples, and we further observed a significant negative enrichment of $P=0.0189/P=0.00141$ in right/left atrial appendage samples of the RNAseq dataset GSE128188 (differential expression results added in **Suppl Table S13**).

Figure 6: Replication of the core gene candidate AF association and NKX2-5 target coexpression in independent datasets.

Published proteomics data (PXD006675) as well as RNAseq data (GSE128188, GTE) generated from human atrial tissue samples were used for replication.

a: Centered and scaled values of the mean mRNA or protein expression in AF ctrls and cases, with stronger effects on protein level. GSEA p-values quantify the negative association of NKX2-5 targets with respect to AF. Sample sizes per column: 69 controls, 14 prevalent AF cases, 69 controls, 14 prevalent AF cases (AFHRI, all right atrial appendage); five controls, five AF cases (GSE128188, both right atrial appendage); five controls, five AF cases (GSE128188, both left atrial appendage); 3 controls, 3 AF cases (PXD006675, both left atrium). A quantitative description of the qualitative results presented in the heatmap can be found in Suppl Table S13-S14 and Table 3.

b: Coexpression of *NKX2-5* with the 13 identified *NKX2-5* transcription factor targets (Pearson's correlation). Quantified is the correlation between *NKX2-5* and its targets on mRNA level for mRNA datasets and the correlation between the *NKX2-5* transcript expression with the target protein concentrations for the AFHRI proteomics (*NKX2-5* not quantified in proteomics). Sample sizes used for the computation of correlations: 102 AFHRI mRNA, 96 AFHRI protein, 372 GTE, ten GSE128188 right, and ten left atrial appendage samples.

AF, atrial fibrillation; Ctrl, control i.e. individuals in sinus rhythm; GSEA, gene set enrichment analysis; GTE, Genotype-Tissue-Expression project; eQTL, expression quantitative trait loci; pQTL, protein quantitative trait loci. *Mutation known to affect cardiovascular phenotypes; **Mutation known to affect arrhythmias; *Differential expression functional impairment for cardiovascular phenotypes; **Differential expression or functional impairment for arrhythmias.

2. Similarly, experimental validation of the *NKX2-5* results would provide compelling support for the manuscript.

Due to the concerns raised by multiple reviewers, we further validated our findings experimentally in the same dataset (added **Suppl Figure S12**) as well as in two independent datasets (added **Figure 6**) with the following results:

1. **rs9481842 - *NKX2-5* trans pQTL (as shown above)**
2. **High correlation of *NKX2-5* protein intensity and estimated *NKX2-5* TF activity:**
 We compared protein abundance to the predicted activity estimated by tissue/cell type specific external annotations and genome-wide transcriptomics data. We observed a Spearman rank correlation of 0.42 (P=0.026, two-sided test, **Suppl Figure S12**).
3. **AF association of 13 identified *NKX2-5* targets in two independent datasets (as shown above)**

4. Coexpression of NKX2-5 with the 13 identified targets (two independent datasets):
We confirmed the regulation of the 13 identified targets by NKX2-5 via coexpression analysis in two independent datasets (**Figure 6b**).

Pearson's correlation of the NKX2-5 transcript with the 13 target transcripts ranged between 0.43 and 0.70 (median: 0.54) in the GTEx (right) atrial appendage tissue. Similarly high correlations were observed in right (0.16-0.81, median: 0.58) and left (0.54-0.87, median: 0.72) atrial appendage samples of the RNAseq dataset GSE128188.

3. The authors should test for cross-mappability potentially explaining their trans-eQTL associations (see PMID: 30613398)

The study on cross-mapping is reporting potential issues in RNA-seq based eQTL studies. In this study, mRNA measurements were performed using micro-arrays. We believe that RNA-seq alignment errors are not an issue in microarray based eQTL studies.

4. The authors selected the covariates to include in their QTL analyses methods based on what covariate combination results in the highest number of significant genes. While that method has been used to select number of latent factors to include, it is not appropriate to not include known confounders (eg. age, sex, disease status) simply because they reduce associations. This could lead to false positives.

In general, we agree with the reviewer, that not including known confounders may lead to false positives. Therefore, we verified that the PEER factors used in the final analysis were correlated with important covariates (**Suppl Figure S2**, reproduced below). We observed high correlations, which support the view that PEER factors might just be a more efficient representation of the (in parts correlated) covariates. We extended **Suppl Figure S2** to now also include ancestry principal components as raised in the next comment.

Figure S2: Correlation of PEER factors with common risk factors of AF and technical covariates.

Pearson correlation of different PEER factors and known risk factors or technical covariates.

a: Transcriptomics analysis: Fibroblast-score and RIN-score highly correlate with PEER factors used in the final *cis* eQTL analysis.

b: Proteomics analysis: fibroblast-score and original sample protein concentration highly correlate with PEER factors used in the final *cis* pQTL analysis.

PEER, probabilistic estimation of expression residuals; QTL, quantitative trait loci; AF, (prevalent) atrial fibrillation; RIN, RNA integrity number; BMI, body mass index; diasBP, diastolic blood pressure; sysBP, systolic blood pressure; HF, heart failure; MI, myocardial infarction; CRP, C-reactive protein; NT-proBNP, N-terminal prohormone of brain natriuretic peptide.

As comment 4 and 5 both concern the inclusion of additional covariates, we would like to address these together in the following.

To evaluate how QTL results differ for different sets of covariates, we reran the analyses. Next to PEER factors, we included three ancestry principle components and age, sex, BMI, disease status and the fibroblast-score as covariates.

In general, including covariates other than the PEER factors reduced the number of detected QTLs (updated **Suppl Figure S1**). The next figure shows that QTL effect sizes correlate extremely well when comparing QTLs derived with PEER factors only as covariates as compared to adding age, sex, BMI, disease status, fibroblast-score and three ancestry principle components (**Figure R4**).

Figure S1: *Cis* QTL analysis results for different covariate sets and number of PEER factors.

PEER analysis was performed to account for unknown variation in the data. Displayed are the number of genes with at least one QTL variant with a $FDR < 0.05$ for different combinations of normalization, number of PEER factors used in the regression and additional covariates. Black dots mark the chosen number of PEER factors and covariates as the maximal number of discovered QTL genes at a $FDR < 0.05$.

QTL, quantitative trait loci; FDR, false discovery rate; PEER, probabilistic estimation of expression residuals; eQTL, expression quantitative trait loci; pQTL, protein quantitative trait loci; res eQTL, residual expression quantitative trait loci; res pQTL, residual protein quantitative trait loci; ratioQTL, ratio quantitative trait loci.

Pearson correlation of effect sizes for all SNP-mRNA pairs, that were significant with any of the sets of covariates ranged between 95.4% and 98.8%. Storey's q-value method estimates replication rates of over 99%. The following **Figure R4** shows ALL SNP-gene pairs that were significant for any of the covariate sets.

Similar observations were made for proteins. Pearson correlation of effect sizes for all SNP-protein pairs, that were significant with any of the sets of covariates ranged between 95.6% and 98.0%. Storey's q-value method estimated replication rates of over 99%. The following plots show ALL SNP-gene pairs that were significant for any of the covariate sets.

Figure R4: Correlation between eQTL/pQTL effect sizes computed using different sets of covariates. We further wanted to address the possibility of false positive associations introduced when considering only PEER factors as compared to PEER factors and additional covariates. We compared the replication rate of our eQTLs to GTEx atrial appendage eQTLs as an independent dataset. The following table shows the estimated π_0 (fraction of true null hypotheses) and the corresponding replication rates ($1 - \pi_0$) when using Storey’s qvalue method:

Table R1

Significance cutoff (AFHRI cohort)	Considered associations	Covariate sets	
		PEER factors only	PEER factors, age, sex, BMI, disease status, fibroblast-score and three ancestry PCs
FDR < 0.05	all SNP-gene pairs	0.08 / 92%	0.09 / 91%
FDR < 0.05	top eQTL per gene	0.41 / 59%	0.40 / 60%
$P < 10^{-5}$	all SNP-gene pairs	0.05 / 95%	0.06 / 94%
$P < 10^{-5}$	top eQTL per gene	0.15 / 85%	0.15 / 85%

This clearly shows equivalent replication rates in the GTEx dataset when using only PEER factors as compared to a more comprehensive set of covariates. Therefore we conclude that there is no evidence of higher rates of false positive findings when using the model optimized for the number of discoveries.

5. Similarly, genotype principal components should be included in all QTL analyses to correct for population substructure

We already addressed this in the response to the previous comment, but we still would like to elaborate that there were no signs of confounding by the population structure.

We evaluated the population structure by comparing our cohort to the 1000 genomes samples of all ancestries and to those of European descent only (Figure R5). All samples were of central European ancestry, without any subclusters or spurious relatedness (red crosses at the center of European samples) for both comparisons. The homogeneous nature of the European samples is also reflected in the extremely small eigenvalues compared to those computed for all populations.

Figure R5: Evaluation of the genetic AFHRI cohort population structure compared to the 1000 genomes individuals.

6. A MAF threshold of 0.01 is too low for inclusion in this study of ~100 participants, they should use 0.05.

We agree with the reviewer that caution is required when analyzing associations with SNPs with low minor allele frequencies in a small cohort, as this might lead to observing only very few samples with the homozygous minor allele genotype. Specifically, having only one or two data points for the homozygous minor allele increases the risk that outliers in the expression data, which coincide with rare genotypes lead to false positive findings. Instead of excluding those variants altogether, our strategy to preclude these spurious associations was to recode homozygous minor allele genotypes to the heterozygous genotype if they were observed in less than 3 samples.

Minor concerns:

1. In addition to validation, the authors should make available in a supplemental table the top genes associated with the Afib PRS based on their QTS analyses with their corresponding association statistics.

We thank the reviewer for this suggestion and have added the requested information in **Suppl Table S6** and **S8**.

2. Do the trans-QTL SNPs tested in Table 2 have a corresponding cis-QTL association? These should be reported, if so, and test whether these associations colocalize with the trans associations suggesting a shared causal signal.

All three trans QTL SNPs have no significant cis QTLs, neither in the AFHRI cohort, nor in GTEx heart atrial appendage tissue.

3. Did the authors test for relatedness between individuals in their cohort? If so, how did they handle sample from related individuals?

Our genotype quality control procedure included the evaluation of relatedness. Kinship was estimated and $\hat{p} > 0.19$ was set as a cutoff. No samples (max. value 0.041) exceeded this threshold and therefore no samples had to be dropped due to relatedness.

4. The authors report that 8.2% of SNP-gene associations have both an eQTL and a pQTL. How many of those colocalize, suggesting a shared causal variant between the two signals?

We thank the reviewer for suggesting colocalization analysis. We had previously addressed shared or independent effects based on the regression approach, but we have now extended the manuscript to include a complementary formal colocalization analysis in the following way:

To evaluate if eQTLs and pQTLs colocalize we can either compare signals for the whole cis-range of each gene, or, to allow further finemapping, perform the analysis on independent loci derived by LD clumping (FDR cutoffs 0.05 for the lead SNPs and FDR cutoff 0.8 for the SNPs that are part of each clump).

Coming back to the original question, all SNP-gene pairs with an overlapping significant eQTL and pQTL (FDR<0.05) (642 pairs for 13 genes) were partitioned into 19 clumps (for 12 genes) where the lead SNP was a significant eQTL and pQTL. All but one (lead SNP rs7202898 for gene CDH13) of those regions formally colocalize with a suggested shared causal variant (coloc posterior probability for a shared causal variant >0.5).

We further evaluated how the formal colocalization analysis complemented our regression based approach to define functional QTL categories.

The residual regression approach and the formal colocalization analyses agreed on the functional category in around two thirds of cases, with the colocalization showing a tendency to be less strict as opposed to the residual regression approach. While both methods detected functional QTL the other did not, both consistently confirmed the existence of the three distinct categories.

We updated **Table 1** to include independent loci as described above and extended **Figure 1** as well as the following sections presenting the different functional QTL categories to incorporate these new results.

Table 1: Summary of tested data and discovered *cis* quantitative trait loci.

Significant *cis* QTLs in human heart right atrial appendage tissue for mRNA and protein measurements for FDR <0.05 (according to Benjamini-Hochberg procedure) and P value <1×10⁻⁵. Loci denote the number of independent loci derived by LD-clumping.

QTL, quantitative trait loci; FDR, false discovery rate; LD, linkage disequilibrium; eQTL, expression quantitative trait loci; pQTL, protein quantitative trait loci; ratioQTL, ratio quantitative trait loci; N, sample size.

Results for all available transcriptomics and proteomics measurements:											
	Tested:			FDR<0.05:			P<1×10 ⁻⁵ :				
	SNPs	Pairs	Genes	Pairs	Genes	Loci	Pairs	Genes	Loci	N	
eQTL	4 861 118	56 139 851	16 306	57 403	1 058	1 657	40 267	552	870	75	
pQTL	2 323 504	4 508 654	1 337	4 081	91	139	2 543	45	71	75	
Results only for genes with both transcriptomics and proteomics measurements:											
eQTL	2 249 758	4 198 168	1 243	4 603	124	201	3 218	64	109	75	
pQTL	2 249 758	4 198 168	1 243	3 906	87	133	2 406	42	66	75	
ratioQTL	2 249 758	4 198 168	1 243	563	16	23	575	18	27	66	

Figure 1: Significant *cis* eQTLs, *cis* pQTLs and their overlap.

a: Circular plot of the significant *cis* eQTLs (blue) and pQTLs (purple) at a FDR cutoff of 0.05 (dotted line), plot created using the R package circlize²¹). Considering only genes with both transcriptomics and proteomics measurements, we visualized the overlap of significant eQTLs and pQTLs in the circle center. In total, the lead SNP-gene-pair of 200 QTL clumps in 124 genes had a significant eQTL and 133 loci in 87 genes a significant pQTL. Only 19 lead variants (13 genes) had an eQTL and pQTL for the same gene. The numbers in brackets represent the number of significant SNP-gene pairs.

b: Characterization of overlapping eQTL and pQTL loci. All 19 LD clumps (based on eQTL and pQTL summary statistics) where the lead SNP-gene-pair was a significant eQTL and pQTL were classified as a shared QTL by either our residual regression approach or colocalization analysis.

c: Characterization of eQTL loci without a corresponding pQTL. Only 83 out of 181 LD clumps (based on eQTL and pQTL summary statistics) that had a lead SNP-gene-pair with a significant eQTL but no pQTL were classified as an independent eQTL by either our residual regression approach or colocalization analysis.

d: Characterization of pQTL loci without a corresponding eQTL. Only 42 out of 114 LD clumps (based on eQTL and pQTL summary statistics) that had a lead SNP-gene-pair with a significant pQTL but no eQTL were classified as an independent pQTL by either our residual regression approach or colocalization analysis.

eQTL, expression quantitative trait loci; pQTL, protein quantitative trait loci; QTL, quantitative trait loci; FDR, false discovery rate; LD, linkage disequilibrium; SNP, single nucleotide polymorphism.

5. The authors report 21 genes and 1083 pQTL variants that don't have a corresponding eQTL in their dataset. How many of those have a concordant eQTL in GTEx Atrial Appendage? Also, how many have a missense variant in their credible set that could explain the discrepancy?

For the 1083 independent pQTLs (defined by regression analysis) in 21 genes, 4 genes had a concordant eQTL in GTEx atrial appendage tissue with a p value smaller than 10^{-5} (NIPSNAP1, ANXA5, CRYZ, YBX3), and AAMDC had a concordant eQTL with a p value $<1.9 \times 10^{-5}$ (Figure R6).

Figure R6: Comparison of eQTL/pQTL effect sizes between the AFHRI cohort and GTEx for independent pQTLs.

The first panel shows the pQTL effect sizes of our independent pQTLs compared to the eQTL effect sizes in GTEx Atrial Appendage tissue.

The second panel shows all independent pQTLs with eQTL and pQTL effect sizes in our AFHRI cohort, even if those SNP-gene pairs were not measured in GTEx atrial appendage tissue.

- GTEx eQTL $< 1e-19$ (NIPSNAP1)
- GTEx eQTL $< 1e-18$ (ANXA5)
- GTEx eQTL $< 1e-12$ (CRYZ)
- GTEx eQTL $< 1e-5$ (YBX3)
- GTEx eQTL $< 1e-4$ (AAMDC)
- GTEx eQTL $> 1e-3$
- SNP-gene pair not evaluated in GTEx

There were 9 missense mutations overlapping with independent pQTLs. Three of those were located in peptides that were used for protein identification and quantification: APOH, BAG3 and YBX3. We did follow up analyses to exclude the possibility that the changed peptide sequence created a measurement artefact and that the alternative allele peptide was simply not quantified because it was not included in the reference database. We included the alternative allele amino acid sequences in the database for MS quantification and repeated the analysis. No peptides including the changed amino acid sequence could be detected for either of the three proteins. This is in line with low expression levels of the alternative alleles for all proteins considered here. Especially for APOH, the measured intensities of the peptide

of interest were high enough that an alternative peptide should have been detected if present.

6. Did the authors confirm that the Afib PRS is associated with atrial fibrillation in their cohort?

We quantified how higher (percentile-transformed) PRS values associated with prevalent AFib ($t=-2.0$, $P=0.026$, one-sided test). We further performed logistic regression analysis to compare the effect of different common risk factors for AFib as well as the PRS in our cohort. The PRS showed stronger effects compared to the other risk factors. Findings are also presented in the added **Suppl Figure S8**.

Figure S8: Genome-wide polygenic score adds relevant information in classifying atrial fibrillation disease status.

a: Percentiles of the atrial fibrillation polygenic risk score by disease status ($t=-2.0$, $P=0.026$, one-sided t-test).

b: Logistic regression results for common risk factors of AF. Significant and comparably strong effect for the PRS variable (estimate 4.7, $P=0.028$, two-sided t-test).

In the boxplots, the lower and upper hinges correspond to the first and third quartiles (the 25th and 75th percentiles). The median is denoted by the central line in the box. The upper/lower whisker extends from the hinge to the largest/smallest value no further than $1.5 \cdot \text{IQR}$ from the hinge.

AF, atrial fibrillation; PRS, genome-wide polygenic score; BMI, body mass index; BP, blood pressure; MI, myocardial infarction; CRP, C-reactive protein; NT-proBNP, N-terminal prohormone of brain natriuretic peptide.

7. The authors need to provide further details on their method for prioritizing NKX2-5 targets for trans association testing. The process is a bit unclear from their methods as currently written. Also, what do the colors in Figure 5d represent? It currently reads as though these are z-scores but it's unclear how those were estimated for each genotype.

In the first step, we prioritized the TF NKX2-5 itself as AF trans eQTL using the approach based on the PRS. In a second step, we established that the expression levels of NKX2-5 targets (defined by NKX2-5 binding sites derived from external ChIP-seq and Hi-C data) collectively show concordant expression changes (estimated NKX2-5 activity) with respect to the genotype. Observing this global trend, we next aimed to identify the individual NKX2-5 targets that drive this association, as they represent further downstream genes of the AF SNP rs9481842.

We have updated the method section to make it more comprehensible:

“To prioritize NKX2-5 target genes with most evidence of an association with the AF SNP which is mediated by the TF, we aimed to establish the following properties:

- a) The target has a NKX2-5 binding site (ChIPseq) overlapping with an open chromatin state in the promoter or a promoter interacting region (HiC).

To establish that NKX2-5 mediates the effect of the AF SNP on the target gene transcription we further need to show that:

- b) The transcript expression of the target gene is associated with the SNP genotype and
- c) the association between the SNP and the target transcript expression is vanishing when adjusting for NKX2-5 expression levels.

Finally, the most relevant endpoint, i.e. target protein abundance, should be mediated by the NKX2-5 TF:

- d) The protein expression of the target is significantly and positively correlated with the transcript expression of NKX2-5.

We do so by evaluating the following regression models:

- 1) Association of GWAS SNP with target transcript (*trans* eQTL, N=67, df=63):

$$\text{target transcript} \sim \beta_0 + \beta_1 \cdot \text{SNP} + \beta_2 \cdot \text{fibroblast-score} + \beta_3 \cdot \text{RIN} + \varepsilon$$
- 2) Independent effects of the SNP on target transcript, that are not mediated by the TF transcript (N=67, df=62):

$$\text{target transcript} \sim \beta_0 + \beta_1 \cdot \text{SNP} + \beta_2 \cdot \text{TF transcript} + \beta_3 \cdot \text{fibroblast-score} + \beta_4 \cdot \text{RIN} + \varepsilon$$
- 3) Association of target protein with TF transcript (N=79, df=75):

$$\text{target protein} \sim \beta_0 + \beta_1 \cdot \text{TF transcript} + \beta_2 \cdot \text{fibroblast-score} + \beta_3 \cdot \text{protein conc.} + \varepsilon$$

Additionally, we quantify the corresponding *trans* pQTL for **Suppl Table S16**:

- 4) Association of GWAS SNP with target protein (*trans* pQTL, N=66, df=62):

$$\text{target protein} \sim \beta_0 + \beta_1 \cdot \text{SNP} + \beta_2 \cdot \text{fibroblast-score} + \beta_3 \cdot \text{protein conc.} + \varepsilon$$

For direct binding a) we considered only genes with transcriptomics and proteomics measurements and at least one functional TF BS.

To establish regulation by the SNP, we selected only genes with a significant association of the SNP with the target transcript (concordant effect to the TF expression) in regression model 1) $\beta_1 < 0$, $P(\beta_1) < 0.05$ to ensure b) and additionally checked the vanishing effect when adding the TF transcript model, i.e. for c) we assessed in regression model 2) $P(\beta_1) > 0.2$.

For the remaining candidates, we established d) that target proteins were associated with the TF transcript and performed FDR correction (Benjamini-Hochberg) on those p values. Based on this, all targets with 3) $\text{FDR}(P(\beta_1) < 0.05) < 0.05$ were defined as functional NKX2-5 targets.”

To make our approach more comprehensible, we outlined our strategy to identify functional NKX2-5 targets graphically in **Suppl Figure S13**:

Figure S13: Definition of functional NKX2-5 targets.

Functional NKX2-5 targets were derived in a three step process: first, by the presence of a most likely functional TF BS, second by a likely regulation of target transcript by the SNP through the TF, and finally, we checked for strong regulation of the target protein by the TF.

TF, transcription factor; BS, binding site; SNP, single-nucleotide polymorphism.

The heatmap in **Figure 5** visualizes the regulatory effects of the SNP rs9481842 on the target transcript (three columns for the genotypes), target protein (three columns for the genotype) and finally the disease association with AF (two columns controls and prevalent AF cases). A more quantitative description of the shown differences can be found in **Suppl Table S16** (rs9481842-target trans eQTL, rs9481842-target trans pQTL and AF differential expression).

Figure 5e: NKX2-5 activity regulated by AF GWAS variant rs9481842.

e: Depicted are functional NKX2-5 targets with the number of TF binding sites (column 1), *trans* eQTL strength (columns 2-4), *trans* pQTL strength (columns 5-7) and protein level in AF (columns 8-9). The color scale represents median transcript or protein values per group (=columns). Residuals corrected for fibroblast-score and RIN-score / protein concentration with subsequent normal-quantile normalization per gene were used to calculate the medians per group. A quantitative description of the qualitative results presented in the heatmap can be found in Suppl Table S16 and Table 3.

The color scale represents median transcript or protein values per group (=columns) and target (=rows). Residuals corrected for fibroblast-score and RIN-score / protein concentration with subsequent normal-quantile normalization per gene were used to match the data described by the linear regression. The medians per group (e.g. the median scaled values of the PPIF transcript residuals for all individuals with the TT genotype in the first square of the heatmap) were then presented in the heatmap.

8. Figure 3c top panel: The pQTL estimate is annotated as having a p-value >0.05 according to the color but the 95% confidence intervals do not include 0. How were those calculated?

Figure 3: Overlap of *cis* QTL associations with GWAS hits annotated in the GWAS catalogue.

a: Overview of significant *cis* eQTLs and pQTLs (FDR<0.05) overlapping with different disease traits.
b: Independent pQTL for GWAS hit creatine kinase levels. Shown are the significant *cis* eQTL, pQTL and ratio QTL for the SNP rs1801690 and the gene APOH (FDR<0.05). In the boxplots, the lower and upper hinges correspond to the first and third quartiles (the 25th and 75th percentiles). The median is denoted by the central line in the box. The upper/lower whisker extends from the hinge to the largest/smallest value no further than 1.5·IQR (interquartile range) from the hinge.
c: For three different trait categories (cardiovascular traits, arrhythmias and atrial fibrillation) as well as rheumatoid arthritis as a negative control, the enrichment of GWAS hits at significant *cis* QTLs (FDR<0.05) was evaluated. Enrichments were calculated using Fisher's exact test (two-sided).
QTL, quantitative trait loci; eQTL, expression quantitative trait loci; pQTL, protein quantitative trait loci; CI, confidence interval;

We thank the reviewer for this important remark as we noticed that our axis labels were incorrect! Displayed are odds ratios instead of log odds ratios as written on the axis label. Accordingly, the dashed reference line needs to be at 1 and not 0. Results were computed using Fisher's exact test (two-sided) and the 95% confidence intervals were derived according to the corresponding test statistic in R. The 95% confidence interval for the discussed pQTL estimate with p value >0.05 for the enrichment of cardiovascular traits now overlaps with the dashed line referencing an odds ratio of 1.

9. How was FDR calculated for Table 1? Is this gene level FDR or genome-wide?

FDR was calculated on all cis snp-gene pairs (i.e. genome-wide).

10. There are some discrepancies in the numbers between the main text and tables/methods in several places:

a) GWAS overlap and enrichment section: text mentions 17 AF-related eQTLs and 4 with pQTLs while Suppl Table S4 lists 15 and 3,

AF-related eQTLs/pQTLs included GWAS hits for Atrial fibrillation (15 and 3), Incident atrial fibrillation (1 and 1) and Prevalent atrial fibrillation (1 and 0), which were listed separately in **Suppl Table S4**.

b) Protein section in the Methods reports measuring protein concentrations in 102 samples but intensities quantified for 96,

One sample was excluded due to irregularities in the chromatographic pattern and 5 samples were left atrial appendage tissues that were not considered in this analysis. The corresponding method section was altered accordingly, stating now:

“To measure the protein concentrations of 97 right atrial appendage samples” and “One sample was excluded due to irregularities in the chromatographic pattern.”

c) cis QTL computation in Methods: There appear to be ~100 samples with RNA and protein quantification and 83 with genotype while cisQTLs are computed for 75 only – is that the overlapping set or were there additional exclusions?

There were no additional exclusions, lower numbers indicate indeed the set of overlapping samples between transcriptomics/proteomics/genomics experiments.

Reviewers' Comments:

Reviewer #1:

Remarks to the Author:

Thank you for the opportunity to re-review this manuscript. I do not have additional major concerns.

Reviewer #2:

Remarks to the Author:

All comments raised have been well addressed. I congratulate the authors on this excellent work.

Reviewer #3:

Remarks to the Author:

The authors have adequately addressed most of my comments. However, failure to replicate the cis pQTLs, trans eQTLs, and trans pQTLs remains a concern as does linking genes/pathways with specific AF mechanisms. Prior studies have provided important insights into the role of NKX2-5 and AF.

Reviewer #4:

Remarks to the Author:

In this revised paper, the authors did a good job addressing the reviewer's concerns. Based on their data I am reassured that the selection of covariates for the QTL analyses and population stratification did not negatively impact the reported associations. The replication of the reported trans-QTLs, albeit weak is sufficient for the scope of this paper which is to be hypothesis-generating for future experimental work.

Response to referees

Reviewer comments

Reviewer 3:

The authors have adequately addressed most of my comments. However, failure to replicate the cis pQTLs, trans eQTLs, and trans pQTLs remains a concern as does linking genes/pathways with specific AF mechanisms. Prior studies have provided important insights into the role of NKX2-5 and AF.

Response:

We agree with the reviewer that important insights about the role of NKX2-5 in AF have previously been reported. We apologize, should we have missed additional aspects and we would be grateful for the corresponding references. In the results section we had cited studies linking NKX2-5 to AF, which we deemed most important:

“In order to get more detailed information about complex molecular mechanisms underlying AF, we further analyzed the TF network of NKX2-5 (see Figure 5a) since the TF has already been described in the context of cardiac development,²⁹ AF,^{30–32} and congenital heart diseases.^{33”}

In the discussion we had also highlighted these findings:

“To investigate more complex molecular mechanisms underlying AF, we further analyzed the TF network of NKX2-5, since the TF has been described in the context of heart development and arrhythmias. For instance, Benaglio and colleagues discovered that changes in NKX2-5 binding can contribute to electrocardiographic phenotypes.³² In addition, a NKX2-5 loss-of-function mutation has been reported to be associated with an increased susceptibility to familial AF³⁰ emphasizing its relevance in the pathology of AF. We investigated downstream effects by inferring NKX2-5 TF activity based on target transcript expression and functional binding sites.”

We have now reformulated the discussion to better put these results into perspective:

“To investigate more complex molecular mechanisms underlying AF, we further analyzed the TF network of NKX2-5, since the TF has been described in the context of heart development and arrhythmias. A loss-of-function mutation in NKX2-5 is associated with increased susceptibility to familial AF³⁰ demonstrating its causal role for AF. However, this result still does not provide mechanistic insights. A study of allele specific binding of NKX2-5 identified relevant targets and demonstrated that variants that change NKX2-5 binding at the promoters of target genes contribute to electrocardiographic phenotypes.³² Hence this study established a mechanistic link of NKX2-5 to AF susceptibility at common GWAS variants, which alter cis regulatory elements bound by NKX2-5 at the target genes. In this work, we link the expression of NKX2-5 to an AF associated GWAS variant in trans. Moreover, we investigated downstream effects by integrating the functional data³² with our genotype and expression data. We identified a set of target genes, where the AF associated variant did not alter the cis regulatory elements but the activity of the trans acting TF NKX2-5. Taken together, prior studies demonstrate a causal link of NKX2-5 with AF and suggested that AF associated variants alter cis regulatory elements. Here we showed that in addition trans acting mechanisms are important to link AF associated variants to their downstream target genes.”

References for this section (numbering from previous revision to make the comparison easier):

29. Anderson, D. J. et al. NKX2-5 regulates human cardiomyogenesis via a HEY2 dependent transcriptional network. *Nat. Commun.* 9, 1373 (2018).
30. Huang, R.-T., Xue, S., Xu, Y.-J., Zhou, M. & Yang, Y.-Q. A novel NKX2-5 loss-of-function mutation responsible for familial atrial fibrillation. *Int. J. Mol. Med.* 31, 1119–1126 (2013).
31. Jhaveri, S., Aziz, P. F. & Saarel, E. Expanding the electrical phenotype of NKX2-5 mutations: Ventricular tachycardia, atrial fibrillation, and complete heart block within one family. *Hear. Case Rep.* 4, 530–533 (2018).
32. Benaglio, P. et al. Allele-specific NKX2-5 binding underlies multiple genetic associations with human electrocardiographic traits. *Nat. Genet.* 51, 1506–1517 (2019).
33. Akazawa, H. & Komuro, I. Cardiac transcription factor Csx/Nkx2-5: Its role in cardiac development and diseases. *Pharmacol. Ther.* 107, 252–268 (2005).